# Accelerated Linearized Laplace Approximation for Bayesian Deep Learning

**Zhijie Deng**[1 2], **Feng Zhou**[2 3], **Jun Zhu**[2 4] [*]
[1] Qing Yuan Research Institute, Shanghai Jiao Tong University
[2] Dept. of Comp. Sci. & Tech., BNRist Center, THU-Bosch Joint ML Center, Tsinghua University
[3] Center for Applied Statistics, School of Statistics, Renmin University of China
[4] Pazhou Laboratory (Huangpu), Guangzhou, China
zhijied@sjtu.edu.cn, feng.zhou@ruc.edu.cn, dcszj@tsinghua.edu.cn

## Abstract

Laplace approximation (LA) and its linearized variant (LLA) enable effortless adaptation of pretrained deep neural networks to Bayesian neural networks. The generalized Gauss-Newton (GGN) approximation is typically introduced to improve their tractability. However, LA and LLA are still confronted with non-trivial inefficiency issues and should rely on Kronecker-factored, diagonal, or even last-layer approximate GGN matrices in practical use. These approximations are likely to harm the fidelity of learning outcomes. To tackle this issue, inspired by the connections between LLA and neural tangent kernels (NTKs), we develop a Nyström approximation to NTKs to accelerate LLA. Our method benefits from the capability of popular deep learning libraries for forward mode automatic differentiation, and enjoys reassuring theoretical guarantees. Extensive studies reflect the merits of the proposed method in aspects of both scalability and performance. Our method can even scale up to architectures like vision transformers. We also offer valuable ablation studies to diagnose our method. Code is available at https://github.com/thudzj/ELLA.

## 1 Introduction

Deep neural networks (DNNs) excel at modeling deterministic relationships and have become *de facto* solutions for diverse pattern recognition problems [18, 54]. However, DNNs fall short in reasoning about model uncertainty [1] and suffer from poor calibration [17]. These issues are intolerable in risk-sensitive scenarios like self-driving [27], healthcare [34], finance [25], etc.

Bayesian Neural Networks (BNNs) have emerged as effective prescriptions to these pathologies [37, 21, 44, 16]. They usually proceed by estimating the posteriors over high-dimensional NN parameters. Due to some intractable integrals, diverse approximate inference methods have been applied to learning BNNs, spanning variational inference (VI) [1, 36], Markov chain Monte Carlo (MCMC) [55, 4], Laplace approximation (LA) [38, 50, 30], etc.

LA has recently gained unprecedented attention because its *post-hoc* nature nicely suits with the *pretraining-finetuning* fashion in deep learning (DL). LA approximates the posterior with a Gaussian around its maximum, whose mean and covariance are the *maximum a posteriori* (MAP) and the inversion of the Hessian respectively. It is a common practice to approximate the Hessian with the generalized Gauss-Newton (GGN) matrix [40] to make the whole workflow more tractable.

Linearized LA (LLA) [14, 22] applies LA to the first-order approximation of the NN of concern. Immer et al. [22] argue that LLA is more sensible than LA in the presence of GGN approximation;

---

[*]Corresponding author. Majority of work completed while Zhijie Deng was at Tsinghua University.

36th Conference on Neural Information Processing Systems (NeurIPS 2022).

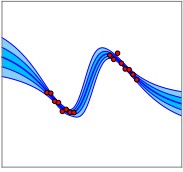 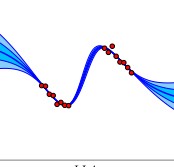 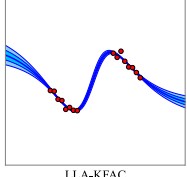 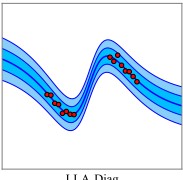 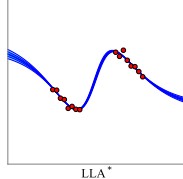

ELLA     LLA     LLA-KFAC     LLA-Diag     LLA*

Figure 1: 1-D regression on $y = \sin 2x + \epsilon, \epsilon \sim \mathcal{N}(0, 0.2)$. Red dots, central blue curves, and shaded regions refer to the training data, mean predictions, and uncertainty respectively. The model is a pretrained multilayer perceptron (MLP) with 3 hidden layers. As shown, the predictive uncertainty of ELLA is on par with or better than the competitors such as LLA with KFAC approximation (LLA-KFAC), LLA with diagonal approximation (LLA-Diag), and last-layer LLA (LLA*).

LLA can perform on par with or better than popular alternatives on various uncertainty quantification (UQ) tasks [14, 6, 7]. The `Laplace` library [6] further substantially advances LLA's applicability, making it a simple and competing baseline for Bayesian DL.

The practical adoption of LA and LLA actually entails further approximations on top of GGN. E.g., when using `Laplace` to process a pretrained ResNet [18], practitioners are recommended to resort to a Kronecker-factored (KFAC) [40] or diagonal approximation of the full GGN matrix for tractability. An orthogonal tactic is to apply LA/LLA to only NNs' last layer [30]. Yet, the approximation errors in these cases can hardly be identified, significantly undermining the fidelity of the learning outcomes.

This paper aims at scaling LLA up to make probabilistic predictions in a more assurable way. We first revisit the inherent connections between Neural Tangent Kernels (NTKs) [24] and LLA [28, 22], and find that, if we can approximate the NTKs with the inner product of some low-dimensional vector representations of the data, LLA can be considerably accelerated. Given this finding, we propose to adapt the Nyström method to approximate the NTKs of multi-output NNs, and advocate leveraging forward mode automatic differentiation (*fwAD*) to efficiently compute the involved Jacobian-vector products (JVPs). The resultant *accElerated LLA (ELLA)* preserves the principal structures of vanilla LLA yet without explicitly computing/storing the costly GGN/Jacobian matrices for the training data. What's more, we theoretically analyze the approximation error between the predictive of ELLA and that of vanilla LLA, and find that it deceases rapidly as the Nyström approximation becomes accurate.

We perform extensive studies to show that ELLA can be a low-cost and effective baseline for Bayesian DL. We first describe how to specify the hyperparameters of ELLA, and use an illustrative regression task to demonstrate the effectiveness of ELLA (see Figure 1). We then experiment on standard image classification benchmarks to exhibit the superiority of ELLA over competing baselines in aspects of both performance and scalability. We further show that ELLA can even scale up to modern architectures like vision transformers (ViTs) [11].

## 2 Background

Consider a learning problem on $\mathcal{D} = (\mathbf{X}, \mathbf{Y}) = \{(\boldsymbol{x}_i, \boldsymbol{y}_i)\}_{i=1}^N$, where $\boldsymbol{x}_i \in \mathcal{X}$ and $\boldsymbol{y}_i \in \mathbb{R}^C$ (e.g., regression) or $\{0, 1\}^C$ (e.g., classification) refer to observations and targets respectively. The advance in machine learning suggests using an NN $g_{\boldsymbol{\theta}}(\cdot) : \mathcal{X} \to \mathbb{R}^C$ with parameters $\boldsymbol{\theta} \in \mathbb{R}^P$ for data fitting. Despite well-performing, regularly trained NNs only capture the most likely interpretation for the data, thus miss the ability to reason about uncertainty and are prone to overfitting and overconfidence.

BNNs [37, 21, 44] characterize model uncertainty by probabilistic principle and can holistically represent all likely interpretations. Typically, BNNs impose a prior $p(\boldsymbol{\theta})$ on NN parameters and chase the Bayesian posterior $p(\boldsymbol{\theta}|\mathcal{D}) = p(\mathcal{D}|\boldsymbol{\theta})p(\boldsymbol{\theta})/p(\mathcal{D})$ where $p(\mathcal{D}|\boldsymbol{\theta}) = \prod_i p(\boldsymbol{y}_i|\boldsymbol{x}_i, \boldsymbol{\theta}) = \prod_i p(\boldsymbol{y}_i|g_{\boldsymbol{\theta}}(\boldsymbol{x}_i))$. Analytical estimation is usually intractable due to NNs' high nonlinearity. Thereby, BNN methods usually find a surrogate of the true posterior $q(\boldsymbol{\theta}) \approx p(\boldsymbol{\theta}|\mathcal{D})$ via approximate inference methods like variational inference (VI) [1, 20, 36, 62, 29], Laplace approximation (LA) [38, 50], Markov chain Monte Carlo (MCMC) [55, 4, 63], particle-optimization based variational inference (POVI) [35], etc.

BNNs predict for new data $\boldsymbol{x}_*$ by posterior predictive $p(\boldsymbol{y}|\boldsymbol{x}_*, \mathcal{D}) = \mathbb{E}_{p(\boldsymbol{\theta}|\mathcal{D})}p(\boldsymbol{y}|\boldsymbol{x}_*, \boldsymbol{\theta}) \approx \mathbb{E}_{q(\boldsymbol{\theta})}p(\boldsymbol{y}|\boldsymbol{x}_*, \boldsymbol{\theta}) \approx \frac{1}{S}\sum_{s=1}^S p(\boldsymbol{y}|g_{\boldsymbol{\theta}_s}(\boldsymbol{x}_*))$ where $\boldsymbol{\theta}_s \sim q(\boldsymbol{\theta})$ are i.i.d. Monte Carlo (MC) samples.

## 2.1 Laplace Approximation and Its Linearized Variant

Typically, LA builds a Gaussian approximate posterior in the form of $q(\boldsymbol{\theta}) = \mathcal{N}(\boldsymbol{\theta}; \hat{\boldsymbol{\theta}}, \boldsymbol{\Sigma})$, where $\hat{\boldsymbol{\theta}}$ denotes the MAP solution, i.e., $\hat{\boldsymbol{\theta}} = \arg\max_{\boldsymbol{\theta}} \log p(\mathcal{D}|\boldsymbol{\theta}) + \log p(\boldsymbol{\theta})$, and $\boldsymbol{\Sigma}$ is the inversion of the Hessian of the negative log posterior w.r.t. parameters, i.e., $\boldsymbol{\Sigma}^{-1} = -\nabla^2_{\boldsymbol{\theta}\boldsymbol{\theta}}(\log p(\mathcal{D}|\boldsymbol{\theta}) + \log p(\boldsymbol{\theta}))|_{\boldsymbol{\theta}=\hat{\boldsymbol{\theta}}}$. Without loss of generality, we base the following discussion on the isotropic Gaussian prior $p(\boldsymbol{\theta}) = \mathcal{N}(\boldsymbol{\theta}; \mathbf{0}, \sigma_0^2 \mathbf{I}_P)$[1], so $-\nabla^2_{\boldsymbol{\theta}\boldsymbol{\theta}} \log p(\boldsymbol{\theta})|_{\boldsymbol{\theta}=\hat{\boldsymbol{\theta}}}$ equals to $\mathbf{I}_P/\sigma_0^2$.

Due to the intractability of the Hessian for NNs with massive parameters, it is a common practice to use the symmetric positive and semi-definite (SPSD) GGN matrix as a workaround, i.e.,

$$\boldsymbol{\Sigma}^{-1} = \sum_i J_{\hat{\boldsymbol{\theta}}}(\boldsymbol{x}_i)^\top \Lambda(\boldsymbol{x}_i, \boldsymbol{y}_i) J_{\hat{\boldsymbol{\theta}}}(\boldsymbol{x}_i) + \mathbf{I}_P/\sigma_0^2, \tag{1}$$

where $J_{\hat{\boldsymbol{\theta}}}(\boldsymbol{x}) \triangleq \nabla_{\boldsymbol{\theta}} g_{\boldsymbol{\theta}}(\boldsymbol{x})|_{\boldsymbol{\theta}=\hat{\boldsymbol{\theta}}}$ and $\Lambda(\boldsymbol{x}, \boldsymbol{y}) \triangleq -\nabla^2_{\boldsymbol{g}\boldsymbol{g}} \log p(\boldsymbol{y}|\boldsymbol{g})|_{\boldsymbol{g}=g_{\hat{\boldsymbol{\theta}}}(\boldsymbol{x})}$.

When concatenating $\{J_{\hat{\boldsymbol{\theta}}}(\boldsymbol{x}_i) \in \mathbb{R}^{C \times P}\}_{i=1}^N$ as a big matrix $\mathbf{J}_{\hat{\boldsymbol{\theta}}, \mathbf{X}} \in \mathbb{R}^{NC \times P}$ and organizing $\{\Lambda(\boldsymbol{x}_i, \boldsymbol{y}_i) \in \mathbb{R}^{C \times C}\}_{i=1}^N$ as a block-diagonal matrix $\boldsymbol{\Lambda}_{\mathbf{X}, \mathbf{Y}} \in \mathbb{R}^{NC \times NC}$, we have:

$$\boldsymbol{\Sigma} = \left[ \mathbf{J}_{\hat{\boldsymbol{\theta}}, \mathbf{X}}^\top \boldsymbol{\Lambda}_{\mathbf{X}, \mathbf{Y}} \mathbf{J}_{\hat{\boldsymbol{\theta}}, \mathbf{X}} + \mathbf{I}_P/\sigma_0^2 \right]^{-1}. \tag{2}$$

Yet, LA suffers from underfitting [33]. This is probably because GGN approximation implicitly turns the original model $g_{\boldsymbol{\theta}}(\boldsymbol{x})$ into a linear one $g_{\boldsymbol{\theta}}^{\text{lin}}(\boldsymbol{x}) = g_{\hat{\boldsymbol{\theta}}}(\boldsymbol{x}) + J_{\hat{\boldsymbol{\theta}}}(\boldsymbol{x})(\boldsymbol{\theta} - \hat{\boldsymbol{\theta}})$ but $g_{\boldsymbol{\theta}}(\boldsymbol{x})$ is still leveraged to make prediction. I.e., there exists a *shift* between posterior inference and prediction [22]. To mitigate this issue, a proposal is to predict with $g_{\boldsymbol{\theta}}^{\text{lin}}(\boldsymbol{x})$, giving rise to linearized LA (LLA) [14, 28, 22].

LA and LLA nicely fit the *pretraining-finetuning* fashion in DL – they can be *post-hoc* applied to pretrained models where only the GGN matrices require to be estimated and further inverted. By LA and LLA, practitioners can adapt off-the-shelf high performing DNNs to BNNs easily.

LLA has revealed strong results on diverse UQ problems [14, 6, 7]. The `Laplace` library [6] further advances LLA's applicability, and evidences LLA is competitive to popular alternatives [63, 32, 39].

**Scalability issue** The GGN matrix of size $P \times P$ is still unamenable in modern DL scenarios, so further approximations sparsifying it are always introduced. The diagonal and KFAC [40] approximations are commonly adopted ones [50, 62], where only a diagonal or block-diagonal structure of the original GNN matrix is preserved. An orthogonal tactic is to concern only a subspace of the high-dimensional parameter space (e.g., the parameter space of the last layer [30]) to reduce the scale of the GGN matrix. However, these strategies sacrifice the fidelity of the learning outcomes as the approximation errors in these cases can hardly be theoretically measured. To this end, we develop *accElerated Linearized Laplace Approximation (ELLA)* to push the limit of LLA in a more assurable way.

## 3 Methodology

In this section, we first revisit the relation of LLA to Gaussian processes (GPs) and Neural Tangent Kernels (NTKs) [24]. After that, we reveal how to accelerate LLA by kernel approximation. Based on these findings, we develop an efficient implementation of ELLA using the Nyström method [58].

### 3.1 The Gaussian Process View of LLA

Integrating $q(\boldsymbol{\theta}) = \mathcal{N}(\boldsymbol{\theta}; \hat{\boldsymbol{\theta}}, \boldsymbol{\Sigma})$ with the linear model $g_{\boldsymbol{\theta}}^{\text{lin}}(\boldsymbol{x})$ actually gives rise to a function-space approximate posterior [14, 28, 22] in the form of $q(f) = \mathcal{GP}(f|g_{\hat{\boldsymbol{\theta}}}(\boldsymbol{x}), \kappa_{\text{LLA}}(\boldsymbol{x}, \boldsymbol{x}'))$ with

$$\kappa_{\text{LLA}}(\boldsymbol{x}, \boldsymbol{x}') \triangleq J_{\hat{\boldsymbol{\theta}}}(\boldsymbol{x}) \boldsymbol{\Sigma} J_{\hat{\boldsymbol{\theta}}}(\boldsymbol{x}')^\top. \tag{3}$$

By Woodbury matrix identity [60], we have:

$$\boldsymbol{\Sigma} = \left[ \mathbf{J}_{\hat{\boldsymbol{\theta}}, \mathbf{X}}^\top \boldsymbol{\Lambda}_{\mathbf{X}, \mathbf{Y}} \mathbf{J}_{\hat{\boldsymbol{\theta}}, \mathbf{X}} + \mathbf{I}_P/\sigma_0^2 \right]^{-1} = \sigma_0^2 \left( \mathbf{I}_P - \mathbf{J}_{\hat{\boldsymbol{\theta}}, \mathbf{X}}^\top [\boldsymbol{\Lambda}_{\mathbf{X}, \mathbf{Y}}^{-1}/\sigma_0^2 + \mathbf{J}_{\hat{\boldsymbol{\theta}}, \mathbf{X}} \mathbf{J}_{\hat{\boldsymbol{\theta}}, \mathbf{X}}^\top]^{-1} \mathbf{J}_{\hat{\boldsymbol{\theta}}, \mathbf{X}} \right). \tag{4}$$

It follows that

$$\kappa_{\text{LLA}}(\boldsymbol{x}, \boldsymbol{x}') = \sigma_0^2 \left( \kappa_{\text{NTK}}(\boldsymbol{x}, \boldsymbol{x}') - \kappa_{\text{NTK}}(\boldsymbol{x}, \mathbf{X})[\boldsymbol{\Lambda}_{\mathbf{X}, \mathbf{Y}}^{-1}/\sigma_0^2 + \kappa_{\text{NTK}}(\mathbf{X}, \mathbf{X})]^{-1} \kappa_{\text{NTK}}(\mathbf{X}, \boldsymbol{x}') \right), \tag{5}$$

---

[1] $\mathbf{I}_P$ refers to the identity matrix of size $P \times P$.

where $\kappa_{\mathrm{NTK}}(\boldsymbol{x}, \boldsymbol{x}') \triangleq J_{\hat{\boldsymbol{\theta}}}(\boldsymbol{x}) J_{\hat{\boldsymbol{\theta}}}(\boldsymbol{x}')^{\top}$ denotes the neural tangent kernel (NTK) [24] corresponding to $g_{\hat{\boldsymbol{\theta}}}(\boldsymbol{x})$. Note that $\kappa_{\mathrm{NTK}}(\boldsymbol{x}, \boldsymbol{x}')$ is a *matrix-valued* kernel, with values in the space of $C \times C$ matrices.

The main challenge then turns into the computation and inversion of the gram matrix $\kappa_{\mathrm{NTK}}(\mathbf{X}, \mathbf{X})$ of size $NC \times NC$. When either $N$ or $C$ is large, the estimation of $\kappa_{\mathrm{LLA}}$ still suffers from inefficiency issue. To address this, existing work [22] assumes independence among the $C$ output dimensions to cast $q(f)$ into $C$ independent GPs following [51], and randomly subsample $M \ll N$ data points to form a cheap substitute for the original gram matrix. Despite effective in some cases, these approximations are heuristic, lacking a clear theoretical foundation.

## 3.2 Scale Up LLA by Kernel Approximation

We show that if we can approximate $\kappa_{\mathrm{NTK}}(\boldsymbol{x}, \boldsymbol{x}')$ with the inner product of some explicit $C \times K$-dimensional representations of the data, i.e., $\kappa_{\mathrm{NTK}}(\boldsymbol{x}, \boldsymbol{x}') \approx \varphi(\boldsymbol{x}) \varphi(\boldsymbol{x}')^{\top}$ with $\varphi : \mathcal{X} \to \mathbb{R}^{C \times K}$, the scalability of the whole workflow can be unleashed.

Concretely, letting $\boldsymbol{\varphi}_{\mathbf{X}} \in \mathbb{R}^{NC \times K}$ be the concatenation of $\{\varphi(\boldsymbol{x}_i) \in \mathbb{R}^{C \times K}\}_{i=1}^{N}$, we have (detailed derivation in Appendix A.2; see also [9])

$$\kappa_{\mathrm{LLA}}(\boldsymbol{x}, \boldsymbol{x}') \approx \sigma_0^2 \Big( \varphi(\boldsymbol{x}) \varphi(\boldsymbol{x}')^{\top} - \varphi(\boldsymbol{x}) \boldsymbol{\varphi}_{\mathbf{X}}^{\top} \Big[ \boldsymbol{\Lambda}_{\mathbf{X}, \mathbf{Y}}^{-1} / \sigma_0^2 + \boldsymbol{\varphi}_{\mathbf{X}} \boldsymbol{\varphi}_{\mathbf{X}}^{\top} \Big]^{-1} \boldsymbol{\varphi}_{\mathbf{X}} \varphi(\boldsymbol{x}')^{\top} \Big)$$

$$= \varphi(\boldsymbol{x}) \underbrace{\Big[ \sum_i \varphi(\boldsymbol{x}_i)^{\top} \Lambda(\boldsymbol{x}_i, \boldsymbol{y}_i) \varphi(\boldsymbol{x}_i) + \mathbf{I}_K / \sigma_0^2 \Big]^{-1}}_{\mathbf{G}} \varphi(\boldsymbol{x}')^{\top} \triangleq \kappa_{\mathrm{ELLA}}(\boldsymbol{x}, \boldsymbol{x}'). \tag{6}$$

The matrix $\mathbf{G} \in \mathbb{R}^{K \times K}$ possesses a similar formula to the $\boldsymbol{\Sigma}$ in the original $\kappa_{\mathrm{LLA}}$ in Equation (3), yet much smaller. Once having $\varphi$, it is only required to perform one forward pass of $g$ and $\varphi$ for each training data to estimate $\mathbf{G}$. When $K$ is reasonably small, e.g., $< 100$, it is cheap to invert $\mathbf{G}$.

## 3.3 Approximate NTKs via the Nyström Method

Typical kernel approximation means include Nyström method [45, 58] and random features-based methods [48, 49, 61, 43, 15, 9]. Given that the latter routinely relies on a relatively large number of random features to gain a faithful approximation (which means a large $K$), we suggest leveraging the Nyström method, which captures only several principal components of the kernel, to build $\varphi$.

To comfortably apply the Nyström method, we first rewrite $\kappa_{\mathrm{NTK}}(\boldsymbol{x}, \boldsymbol{x}')$ as a *scalar-valued* kernel $\kappa_{\mathrm{NTK}}\left((\boldsymbol{x}, i), (\boldsymbol{x}', i')\right) = J_{\hat{\boldsymbol{\theta}}}(\boldsymbol{x}, i) J_{\hat{\boldsymbol{\theta}}}(\boldsymbol{x}', i')^{\top}$ where $J_{\hat{\boldsymbol{\theta}}}(\boldsymbol{x}, i) \triangleq \nabla_{\boldsymbol{\theta}} [g_{\boldsymbol{\theta}}(\boldsymbol{x})]^{(i)} |_{\boldsymbol{\theta} = \hat{\boldsymbol{\theta}}} : \mathcal{X} \times [C] \to \mathbb{R}^{1 \times P2}$ computes the gradient of $i$-th output w.r.t. parameters.

By Mercer's theorem [42],

$$\kappa_{\mathrm{NTK}}\left((\boldsymbol{x}, i), (\boldsymbol{x}', i')\right) = \sum_{k \geq 1} \mu_k \psi_k(\boldsymbol{x}, i) \psi_k(\boldsymbol{x}', i'), \tag{7}$$

where $\psi_k \in L^2(\mathcal{X} \times [C], q)$[3] refer to the eigenfunctions of the NTK w.r.t. some probability measure $q$ with $\mu_k \geq 0$ as the associated eigenvalues. Based on this, the Nyström method discovers the top-$K$ eigenvalues as well as the corresponding eigenfunctions for kernel approximation.

By definition, the eigenfunctions can represent the spectral information of the kernel:

$$\int \kappa_{\mathrm{NTK}}\left((\boldsymbol{x}, i), (\boldsymbol{x}', i')\right) \psi_k(\boldsymbol{x}', i') q(\boldsymbol{x}', i') = \mu_k \psi_k(\boldsymbol{x}, i), \forall k \geq 1, \tag{8}$$

while being orthonormal under $q$:

$$\int \psi_k(\boldsymbol{x}, i) \psi_{k'}(\boldsymbol{x}, i) q(\boldsymbol{x}, i) = \mathbb{1}[k = k'], \forall k, k' \geq 1. \tag{9}$$

In our case, $q(\boldsymbol{x}, i)$ can be factorized as the product of the data distribution and a uniform distribution over $\{1, ..., C\}$, which can be trivially sampled from. The Nyström method draws $M$ ($M \geq K$) i.i.d. samples $\tilde{\mathbf{X}} = \{(\boldsymbol{x}_1, i_1), ..., (\boldsymbol{x}_M, i_M)\}$ from $q$ to approximate the integration in Equation (8):

$$\frac{1}{M} \sum_{m=1}^{M} \kappa_{\mathrm{NTK}}\left((\boldsymbol{x}, i), (\boldsymbol{x}_m, i_m)\right) \psi_k(\boldsymbol{x}_m, i_m) = \mu_k \psi_k(\boldsymbol{x}, i), \forall k \in [K]. \tag{10}$$

---

[2]We use superscript to index a multi-dimensional tensor and $[C]$ represents set of integers from 1 to $C$.

[3]$L^2(, q)$ denotes the space of square-integrable functions w.r.t. $q$.

Applying this equation to these samples gives rise to:

$$\frac{1}{M} \sum_{m=1}^{M} \kappa_{\text{NTK}} \left( (\boldsymbol{x}_{m'}, i_{m'}), (\boldsymbol{x}_m, i_m) \right) \psi_k(\boldsymbol{x}_m, i_m) = \mu_k \psi_k(\boldsymbol{x}_{m'}, i_{m'}), \forall k \in [K], m' \in [M], \quad (11)$$

then we arrive at

$$\frac{1}{M} \mathbf{K} \boldsymbol{\psi}_k \approx \mu_k \boldsymbol{\psi}_k, k \in [K], \quad (12)$$

where $\mathbf{K} = \mathbf{J}_{\hat{\boldsymbol{\theta}}, \tilde{\mathbf{X}}} \mathbf{J}_{\hat{\boldsymbol{\theta}}, \tilde{\mathbf{X}}}^{\top}$, with $\mathbf{J}_{\hat{\boldsymbol{\theta}}, \tilde{\mathbf{X}}} \in \mathbb{R}^{M \times P}$ as the concatenation of $\{J_{\hat{\boldsymbol{\theta}}}(\boldsymbol{x}_m, i_m) \in \mathbb{R}^{1 \times P}\}_{m=1}^{M}$, and $\boldsymbol{\psi}_k = [\psi_k(\boldsymbol{x}_1, i_1), ..., \psi_k(\boldsymbol{x}_M, i_M)]^{\top}$. This implies a scaled eigendecomposition problem of $\mathbf{K}$.

We compute the top-$K$ eigenvalues $\lambda_1, ..., \lambda_K$ of matrix $\mathbf{K}$ and record the corresponding *orthonormal* eigenvectors $\boldsymbol{u}_1, ..., \boldsymbol{u}_K$. Given the constraint in Equation (9), it is easy to see that:

$$\mu_k \approx \frac{\lambda_k}{M}, \text{ and } \psi_k(\boldsymbol{x}_m, i_m) \approx \sqrt{M} \boldsymbol{u}_k^{(m)}, m \in [M]. \quad (13)$$

Combining Equation (13) and (10) yields the Nyström approximation of the top-$K$ eigenfunctions:

$$\hat{\psi}_k(\boldsymbol{x}, i) = \frac{\sqrt{M}}{\lambda_k} \sum_{m=1}^{M} \boldsymbol{u}_k^{(m)} \kappa_{\text{NTK}} \left( (\boldsymbol{x}, i), (\boldsymbol{x}_m, i_m) \right) = \frac{\sqrt{M}}{\lambda_k} J_{\hat{\boldsymbol{\theta}}}(\boldsymbol{x}, i) \mathbf{J}_{\hat{\boldsymbol{\theta}}, \tilde{\mathbf{X}}}^{\top} \boldsymbol{u}_k. \quad (14)$$

Then, the mapping $\varphi$ which satisfied $\kappa_{\text{NTK}}(\boldsymbol{x}, \boldsymbol{x}') \approx \varphi(\boldsymbol{x}) \varphi(\boldsymbol{x}')^{\top}$ can be realized as:

$$[\varphi(\boldsymbol{x})]^{(i,k)} = \hat{\psi}_k(\boldsymbol{x}, i) \sqrt{\mu_k} = J_{\hat{\boldsymbol{\theta}}}(\boldsymbol{x}, i) \mathbf{J}_{\hat{\boldsymbol{\theta}}, \tilde{\mathbf{X}}}^{\top} \boldsymbol{u}_k / \sqrt{\lambda_k}, \quad (15)$$

$$\Rightarrow \varphi(\boldsymbol{x}) = [J_{\hat{\boldsymbol{\theta}}}(\boldsymbol{x}) \boldsymbol{v}_1, ..., J_{\hat{\boldsymbol{\theta}}}(\boldsymbol{x}) \boldsymbol{v}_K] \text{ with } \boldsymbol{v}_k = \mathbf{J}_{\hat{\boldsymbol{\theta}}, \tilde{\mathbf{X}}}^{\top} \boldsymbol{u}_k / \sqrt{\lambda_k}. \quad (16)$$

**Connection to sparse approximations of GPs** A recent work [57] has shown that sparse variational Gaussian processes (SVGP) [53, 3] is algebraically equivalent to the Nyström approximation because the Evidence Lower BOund (ELBO) of the former encompasses the learning objective of the latter. That's to say, ELLA implicitly deploys a sparse approximation to the original GP predictive of LLA.

**Discussion** In this work, the developed Nyström approximation to NTKs is mainly used for accelerating the computation of the predictive covariance of LLA, while it can also be easily applied to future work on the practical application of NTKs.

## 3.4 Implementation

As shown, the estimation of $\varphi$ on a data point $\boldsymbol{x}$ degenerates as $K$ JVPs, which can be accomplished by invoking *fwAD* for $K$ times:

$$\begin{pmatrix} \boldsymbol{x}, \dfrac{\hat{\boldsymbol{\theta}}}{\boldsymbol{v}_k} \end{pmatrix} \xrightarrow{\quad \text{forward} \quad} \dfrac{g_{\hat{\boldsymbol{\theta}}}(\boldsymbol{x})}{J_{\hat{\boldsymbol{\theta}}}(\boldsymbol{x}) \boldsymbol{v}_k}, k \in [K], \quad (17)$$

where the model output $g_{\hat{\boldsymbol{\theta}}}(\boldsymbol{x})$ and the JVP $J_{\hat{\boldsymbol{\theta}}}(\boldsymbol{x}) \boldsymbol{v}_k$ are simultaneously computed in *one single forward pass*. Prevalent DL libraries like PyTorch [47] and Jax [2] have already been armed with the capability for *fwAD*. Algorithm 2 details the procedure of building $\varphi$ in a PyTorch-like style.

With that, we can trivially compute the posterior $q(f)$ according to Equation (6), as detailed in Algorithm 1. The procedures for estimating $\mathbf{G}$ and the posterior can both be batched for acceleration.

**Computation overhead** The estimation of $\varphi$ involves $M$ forward and backward passes of $g_{\hat{\boldsymbol{\theta}}}$ and the eigendecomposition of a matrix of size $M \times M$. After that, we only need to make $\{\boldsymbol{v}_k\}_{k=1}^{K}$ persistent, which amounts to *storing $K$ more NN copies*. Estimating $\mathbf{G}^{-1}$ requires scanning the training set once and inverting a matrix of $K \times K$, which is cheap. The evaluation of $q(f)$ embodies that of $\varphi$, i.e., performing $K$ *forward passes under the scope of fwAD*. This is similar to other BNNs that perform $S$ forward passes with different MC parameter samples to estimate the posterior predictive.

## 4 Theoretical Analysis

In this section, we theoretically analyze the approximation error of $\kappa_{\text{ELLA}}(\boldsymbol{x}, \boldsymbol{x}')$ to $\kappa_{\text{LLA}}(\boldsymbol{x}, \boldsymbol{x}')$.

<table>
<tr><th>Algorithm 1: Build the LLA posterior.</th><th>Algorithm 2: Build $\varphi$.</th></tr>
</table>

```
# g_θ̂: NN pre-trained by MAP; (X,Y):
# training set; C: number of classes
# M,K,σ₀²: hyper-parameters
def estimate_G(φ,X,Y,K,σ₀²):
    G =zeros(K,K)
    for (x,y) in (X,Y):
        g_x,φ_x = φ(x)
        Λ_x,y = hessian(nll(g_x,y),g_x)
        G += φ_xᵀΛ_x,yφ_x
    return G+ eye(K)/σ₀²
def _q_f(φ,G⁻¹,x)
    g_x,φ_x = φ(x)
    κ_x,x = φ_xG⁻¹φ_xᵀ
    return N(g_x,κ_x,x)
φ =build_φ(g_θ̂, X, C, M, K)
G⁻¹ =inv(estimate_G(φ,X,Y,K,σ₀²))
q_f = partial(_q_f,φ,G⁻¹)
```

```
def build_φ(g_θ̂,X,C,M,K):
    def _φ(g_θ̂,C,{v_k}ᴷₖ₌₁,x):
        φ_x = zeros(C,K)
        for k in range(K):
            with fwAD.enable():
                g_x,jvp = g_(θ̂,v_k)(x)
            φ_x[:,k] = jvp
        return g_x,φ_x
    J_θ̂,X̃ =zeros(M,dim(θ̂))
    for m in range(M):
        x_m = uniform_sample(X)
        i_m = uniform_sample([C])
        J_θ̂,X̃[m]=grad(g_θ̂(x_m)[i_m],θ̂)
    {λ_k,u_k}ᴷₖ₌₁ = eig(J_θ̂,X̃J_θ̂,X̃ᵀ,top = K)
    for k in range(K):
        v_k = J_θ̂,X̃ᵀu_k/√λ_k
    return partial(_φ,g_θ̂,C,{v_k}ᴷₖ₌₁)
```

To reduce unnecessary complexity, this section assumes using $M = K$ i.i.d. MC samples when performing the Nyström method. Then, $\kappa_{\text{ELLA}}$ can be reformulated as (details deferred to Appendix A.3)

$$\kappa_{\text{ELLA}}(\boldsymbol{x},\boldsymbol{x}') = J_{\hat{\boldsymbol{\theta}}}(\boldsymbol{x})\underbrace{\mathbf{J}_{\hat{\boldsymbol{\theta}},\tilde{\mathbf{X}}}^{\top}\Big[\mathbf{J}_{\hat{\boldsymbol{\theta}},\tilde{\mathbf{X}}}\mathbf{J}_{\hat{\boldsymbol{\theta}},\mathbf{X}}^{\top}\boldsymbol{\Lambda}_{\mathbf{X},\mathbf{Y}}\mathbf{J}_{\hat{\boldsymbol{\theta}},\mathbf{X}}\mathbf{J}_{\hat{\boldsymbol{\theta}},\tilde{\mathbf{X}}}^{\top} + \mathbf{J}_{\hat{\boldsymbol{\theta}},\tilde{\mathbf{X}}}\mathbf{J}_{\hat{\boldsymbol{\theta}},\tilde{\mathbf{X}}}^{\top}/\sigma_0^2\Big]^{-1}\mathbf{J}_{\hat{\boldsymbol{\theta}},\tilde{\mathbf{X}}}}_{\boldsymbol{\Sigma}'} J_{\hat{\boldsymbol{\theta}}}(\boldsymbol{x})^{\top}. \qquad (18)$$

Thereby, the gap between $\kappa_{\text{ELLA}}(\boldsymbol{x},\boldsymbol{x}')$ and $\kappa_{\text{LLA}}(\boldsymbol{x},\boldsymbol{x}')$ can be upper bounded by $\mathcal{E} = \|\boldsymbol{\Sigma}' - \boldsymbol{\Sigma}\|$, where $\|\cdot\|$ represents the matrix 2-norm (i.e., the spectral norm). In typical cases, $P \gg M$, thus with high probability $\mathbf{K} = \mathbf{J}_{\hat{\boldsymbol{\theta}},\tilde{\mathbf{X}}}\mathbf{J}_{\hat{\boldsymbol{\theta}},\tilde{\mathbf{X}}}^{\top} \succ 0$. We present an upper bound of $\mathcal{E}$ as follows.

**Theorem 1** (Proof in Appendix A.4). *Let $c_\Lambda$ be a finite constant associated with $\Lambda$, and $\mathcal{E}'$ the error of Nyström approximation $\|\mathbf{J}_{\hat{\boldsymbol{\theta}},\mathbf{X}}\mathbf{J}_{\hat{\boldsymbol{\theta}},\tilde{\mathbf{X}}}^{\top}(\mathbf{J}_{\hat{\boldsymbol{\theta}},\tilde{\mathbf{X}}}\mathbf{J}_{\hat{\boldsymbol{\theta}},\tilde{\mathbf{X}}}^{\top})^{-1}\mathbf{J}_{\hat{\boldsymbol{\theta}},\tilde{\mathbf{X}}}\mathbf{J}_{\hat{\boldsymbol{\theta}},\mathbf{X}}^{\top} - \mathbf{J}_{\hat{\boldsymbol{\theta}},\mathbf{X}}\mathbf{J}_{\hat{\boldsymbol{\theta}},\mathbf{X}}^{\top}\|$. It holds that*

$$\mathcal{E} \le \sigma_0^4 c_\Lambda \mathcal{E}' + \sigma_0^2.$$

$\mathcal{E}'$ has been extensively analyzed by pioneering work [12, 5, 26], and we simply adapt the results developed by Drineas and Mahoney [12] to our case. We denote the maximum diagonal entry and the $M + 1$-th largest eigenvalue of $\mathbf{J}_{\hat{\boldsymbol{\theta}},\mathbf{X}}\mathbf{J}_{\hat{\boldsymbol{\theta}},\mathbf{X}}^{\top}$ by $c_\kappa$ and $\tilde{\lambda}_{M+1}$ respectively.

**Theorem 2** (Error bound of Nyström approximation). *With probability at least $1 - \delta$, it holds that:*

$$\mathcal{E}' \le \tilde{\lambda}_{M+1} + \frac{NC}{\sqrt{M}}c_\kappa(2 + \log\frac{1}{\delta}).$$

Plugging this back to $\mathcal{E}$, we arrive at the corollary below.

**Corollary 1.** *With probability at least $1 - \delta$, the following bound exists:*

$$\mathcal{E} \le \sigma_0^4 c_\Lambda(\tilde{\lambda}_{M+1} + \frac{NC}{\sqrt{M}}c_\kappa(2 + \log\frac{1}{\delta})) + \sigma_0^2.$$

As desired, the upper bound of $\mathcal{E}$ decreases along with the growing of the number of MC samples $M$.

## 5 Experiments

We first discuss how to specify the hyperparameters of ELLA, then expose an interesting finding. After that, we compare ELLA to competing baselines to evidence its merits in efficacy and scalability.

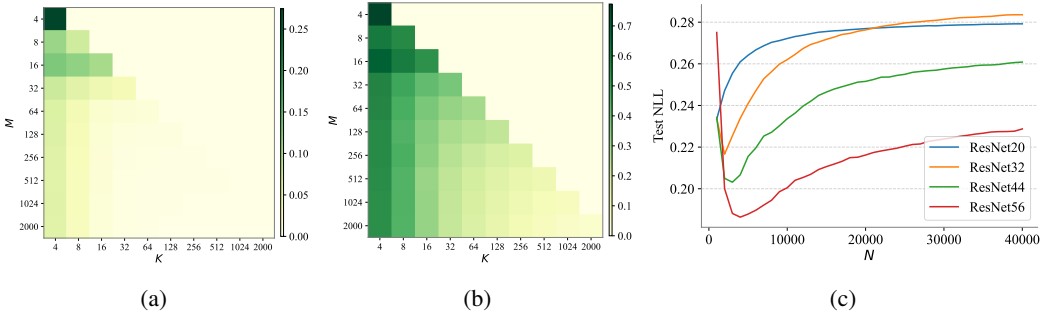

(a)          (b)          (c)

Figure 2: (a)-(b): The approximation errors $\epsilon_{\text{Nyström}}$ and $\epsilon_{\text{ELLA}}$ vary w.r.t. $M$ and $K$ ($M \geq K$). (c): The test NLL of ELLA on CIFAR-10 varies w.r.t. the number of training data $N$.

## 5.1 Specification of Hyperparameters

Given an NN $g_{\hat{\boldsymbol{\theta}}}$ pretrained by MAP, we need to specify the prior variance $\sigma_0^2$, the number of MC samples in Nyström method $M$, and the number of remaining eigenpairs $K$ before applying ELLA. We simply set $\sigma_0^2$ to $\frac{1}{N\gamma}$ with $\gamma$ as the weight decay coefficient used for pretraining according to [10].

We perform an empirical study to inspect how $M$ and $K$ affect the quality of the Nyström approximation and the ELLA GP covariance. We take 2000 MNIST images as training set $\mathbf{X}$, and 256 others as validation set $\mathbf{X}_{\text{val}}$. The architecture contains 2 convolutions and a linear head. Batch normalization [23] and ReLU activation are used. The number of parameters $P$ is $29,034$. Larger architectures or larger $\mathbf{X}$ will render the exact evaluation of $\kappa_{\text{LLA}}$ unapproachable. We quantify the approximation error of the Nyström method by $\epsilon_{\text{Nyström}} \triangleq \|\mathbf{J}_{\hat{\boldsymbol{\theta}},\mathbf{X}}\mathbf{J}_{\hat{\boldsymbol{\theta}},\tilde{\mathbf{X}}}^{\top}(\mathbf{J}_{\hat{\boldsymbol{\theta}},\tilde{\mathbf{X}}}\mathbf{J}_{\hat{\boldsymbol{\theta}},\tilde{\mathbf{X}}}^{\top})^{-1}\mathbf{J}_{\hat{\boldsymbol{\theta}},\mathbf{X}}\mathbf{J}_{\hat{\boldsymbol{\theta}},\mathbf{X}}^{\top} - \mathbf{J}_{\hat{\boldsymbol{\theta}},\mathbf{X}}\mathbf{J}_{\hat{\boldsymbol{\theta}},\mathbf{X}}^{\top}\|/\|\mathbf{J}_{\hat{\boldsymbol{\theta}},\mathbf{X}}\mathbf{J}_{\hat{\boldsymbol{\theta}},\mathbf{X}}^{\top}\|$, and that from $\kappa_{\text{ELLA}}$ to $\kappa_{\text{LLA}}$ by $\epsilon_{\text{ELLA}} \triangleq \frac{1}{|\mathbf{X}_{\text{val}}|}\sum_{\boldsymbol{x}\in\mathbf{X}_{\text{val}}}\|\kappa_{\text{ELLA}}(\boldsymbol{x},\boldsymbol{x}) - \kappa_{\text{LLA}}(\boldsymbol{x},\boldsymbol{x})\|/\|\kappa_{\text{LLA}}(\boldsymbol{x},\boldsymbol{x})\|$. We vary $M$ from 4 to 2000 and $K$ from 4 to $M$, and report the approximation errors in Figure 2 (a)-(b). We notice that 1) the larger $K$ the better; 2) when fixing $K$, $\epsilon_{\text{Nyström}}$ and $\epsilon_{\text{ELLA}}$ decay as $M$ grows; 3) $\epsilon_{\text{Nyström}}$ decays more rapidly than $\epsilon_{\text{ELLA}}$, echoing Theorem 1. Given that ELLA needs to store $K$ vectors of size $P$, small $K$ is desired for efficiency. $K \in [16, 32]$ seems to be a reasonable choice given Figure 2. Besides, Appendix C.1 includes a direct study on how the test NLL of ELLA varies w.r.t. $M$ and $K$ on CIFAR-10 [31] benchmark. Given these results, we set $M = 2000$ and $K = 20$ in the following experiments unless otherwise stated.

ELLA (or LLA) finds a GP posterior, so predicts with a variant of the aforementioned posterior predictive, formulated as $p(\boldsymbol{y}|\boldsymbol{x}_*, \mathcal{D}) = \mathbb{E}_{p(f|\mathcal{D})}p(\boldsymbol{y}|f(\boldsymbol{x}_*)) \approx \mathbb{E}_{q(f)}p(\boldsymbol{y}|f(\boldsymbol{x}_*)) = \mathbb{E}_{\boldsymbol{f}\sim\mathcal{N}(g_{\hat{\boldsymbol{\theta}}}(\boldsymbol{x}_*),\kappa_{\text{ELLA}}(\boldsymbol{x}_*,\boldsymbol{x}_*))}p(\boldsymbol{y}|\boldsymbol{f})$. In classification tasks, we use $512$ MC samples to approximate the last expectation and it is cheap.

## 5.2 The Overfitting Issue of LLA

Reinspecting Equation (1) and (6), we see, with more training data involved, the covariance in LA, LLA, and ELLA shrinks and the uncertainty dissipates. However, under ubiquitous model misspecification [41], *should the uncertainty vanish that fast?* We perform a study with ResNets [18] on CIFAR-10 to seek an answer. Concretely, we randomly subsample $N$ data from the training set of CIFAR-10, and fit ELLA on them. We depict the test negative log-likelihood (NLL), accuracy, and expected calibration error (ECE) [17] of the deployed ELLA in Figure 2 (c) and Appendix C.2. We also provide the corresponding results of LLA*[4] in Appendix C.3. The *V-shape* NLL curves across settings reflect the *overfitting* issue of ELLA and LLA* (or more generally, LLA). Figure 7 also shows that tuning the prior precision w.r.t. marginal likelihood can alleviate the overfitting of LLA* to some extent, which may be the reason why such an issue has not been reported by previous works. But also of note that tuning the prior precision cannot fully eliminate overfitting.

To address the overfitting issue, we advocate *performing early stop when fitting ELLA/LLA on big data*. Specifically, when iterating over the training data to estimate $\mathbf{G}$ (see Equation (6)), we

---

[4]We experiment on LLA* here due to its higher efficiency than LLA-KFAC and LLA-Diag. LLA* is the default option in the `Laplace` library [6].

Table 1: Comparison on test accuracy (%) ↑, NLL ↓, and ECE ↓ on CIFAR-10. We report the average results over 5 random runs. As the accuracy values of most methods are close, we do not highlight the best.

| Method | ResNet-20 | | | ResNet-32 | | | ResNet-44 | | | ResNet-56 | | |
|---|---|---|---|---|---|---|---|---|---|---|---|---|
| | Acc. | NLL | ECE | Acc. | NLL | ECE | Acc. | NLL | ECE | Acc. | NLL | ECE |
| *ELLA* | 92.5 | 0.233 | **0.009** | 93.5 | **0.215** | **0.008** | 93.9 | **0.204** | **0.007** | 94.4 | **0.187** | **0.007** |
| *MAP* | 92.6 | 0.282 | 0.039 | 93.5 | 0.292 | 0.041 | 94.0 | 0.275 | 0.039 | 94.4 | 0.252 | 0.037 |
| *MFVI-BF* | 92.7 | **0.231** | 0.016 | 93.5 | 0.222 | 0.020 | 93.9 | 0.206 | 0.018 | 94.4 | 0.188 | 0.016 |
| *LLA** | 92.6 | 0.269 | 0.034 | 93.5 | 0.259 | 0.033 | 94.0 | 0.237 | 0.028 | 94.4 | 0.213 | 0.022 |
| *LLA*-KFAC* | 92.6 | 0.271 | 0.035 | 93.5 | 0.260 | 0.033 | 94.0 | 0.232 | 0.028 | 94.4 | 0.202 | 0.024 |
| *LLA-Diag* | 92.2 | 0.728 | 0.404 | 92.7 | 0.755 | 0.430 | 92.8 | 0.778 | 0.445 | 92.9 | 0.843 | 0.480 |
| *LLA-KFAC* | 92.0 | 0.852 | 0.467 | 91.8 | 1.027 | 0.547 | 91.4 | 1.091 | 0.566 | 89.8 | 1.174 | 0.579 |

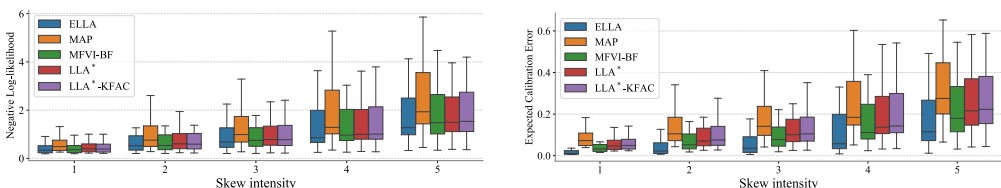

Figure 3: NLL (Left) and ECE (Right) on CIFAR-10 corruptions for models trained with ResNet-56 architecture. Each box corresponds to a summary of the results across 19 types of skew.

continuously record the NLL of the current ELLA posterior on some validation data, and stop when there is a trend of overfitting. If we cannot access a validation set easily, we can apply strong data augmentation to some training data to form a substitute of the validation set. Compared to tuning the prior, early stopping also helps reduce the time cost of processing big data (e.g., ImageNet [8]).

### 5.3 Illustrative Regression

We build a regression problem with $N = 16$ samples from $y = \sin 2x + \epsilon, \epsilon \sim \mathcal{N}(0, 0.2)$ as shown in Figure 1. The model is an MLP with 3 hidden layers and `tanh` activations, and we pretrain it by MAP for 1000 iterations. For ELLA, we set $M = 16$ and $K = 5$ for efficiency. Unless stated otherwise, we use the interfaces of `Laplace` [6] to implement LLA, LLA-KFAC, LLA-Diag, and LLA*. The hyperparameters of the competitors are *equivalent* to those of ELLA except for some dedicated ones like $M$ and $K$. It is clear that ELLA delivers a closer approximation to LLA than LLA-Diag and LLA*. We further quantify the quality of the predictive distributions produced by these approximate LLA methods using certain metrics. Considering that in this case, the predictive distribution for one test datum is a Gaussian distribution, we use the KL divergence between the Gaussians yielded by the approximate LLA method and vanilla LLA as a proxy of the approximation error (averaged over a set of test points). The results are reported in Appendix C.4. As shown, LLA-KFAC comes pretty close to LLA. Yet, LLA seems to *underestimate* in-between uncertainty in this setting, so ELLA seems to be a more reliable (instead of more accurate) approximation than LLA-KFAC. We also highlight the higher scalability of ELLA than LLA-KFAC (see Figure 4(c)), which reflects that ELLA can strike a good trade-off between efficacy and efficiency.

### 5.4 CIFAR-10 Classification

Then, we evaluate ELLA on CIFAR-10 benchmark using ResNet architectures [18]. We obtain pretrained MAP models from the open source community. Apart from MAP, LLA*, LLA-Diag, LLA-KFAC, we further introduce last-layer LLA with KFAC approximation (LLA*-KFAC) and mean-field VI via Bayesian finetuning [10] (MFVI-BF)[5] as baselines. LLA cannot be directly applied as the GGN matrices are huge. These methods all locate in the family of Gaussian approximate posteriors and are all *post-hoc* applied to the pretrained models, so the comparisons will be fair. Regarding the setups, we use $M = 2000$ and $K = 20$ for ELLA;[6] we use 20 MC samples to estimate

---

[5]Flipout [56] trick is deployed for variance-reduced gradient estimation.

[6]Storing 2000 vectors of size $P$ is costly, so we trade time for memory and compute them whenever needed.

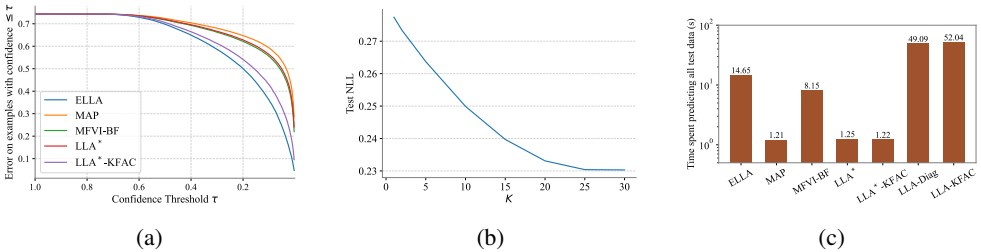

Figure 4: (a) Error versus confidence plots for methods trained on CIFAR-10 and tested on CIFAR-10+SVHN. (b) Test NLL of ELLA varies w.r.t. $K$ on CIFAR-10. (c) Comparison on the wall-clock time used for predicting all CIFAR-10 test data (measured on an NVIDIA A40 GPU). The results are obtained with ResNet-20 architecture.

Table 2: Comparison on test accuracy (%) ↑, NLL ↓, and ECE ↓ on ImageNet. We report the average results over 3 random runs.

| Method | ResNet-18 | | | ResNet-34 | | | ResNet-50 | | |
| --- | --- | --- | --- | --- | --- | --- | --- | --- | --- |
| | Acc. | NLL | ECE | Acc. | NLL | ECE | Acc. | NLL | ECE |
| *ELLA* | 69.8 | 1.243 | **0.015** | 73.3 | 1.072 | **0.018** | **76.2** | 0.948 | **0.018** |
| *MAP* | 69.8 | 1.247 | 0.026 | 73.3 | 1.081 | 0.035 | **76.2** | 0.962 | 0.037 |
| *MFVI-BF* | **70.3** | **1.218** | 0.042 | **73.7** | **1.043** | 0.033 | 76.1 | **0.945** | 0.030 |

the posterior predictive of MFVI-BF (as it incurs 20 NN forward passes), and use 512 ones for the other methods as stated. We have enabled *the tuning of the prior precision* for all LLA baselines, but not for ELLA.

We present the comparison on test accuracy, NLL, and ECE in Table 1. As shown, ELLA exhibits superior NLL and ECE across settings. MFVI-BF also gives good NLL. LLA* and LLA*-KFAC can improve the uncertainty and calibration of MAP, yet underperforming ELLA. LLA-Diag and LLA-KFAC fail for unclear reasons (also reported by [9]), we thus exclude them from the following studies.

We then examine the models on the widely used out-of-distribution (OOD) generalization/robustness benchmark CIFAR-10 corruptions [19] and report the results in Figure 3 and Appendix C.5. It is prominent that ELLA surpasses the baselines in aspects of NLL and ECE at various levels of skew, implying its ability to make conservative predictions for OOD inputs.

We further evaluate the models on the combination of the CIFAR-10 test data and the OOD SVHN test data. The predictions on SVHN are all regarded as wrong due to label shift. We plot the average error rate of the samples with $\leq \tau$ ($\tau \in [0, 1]$) confidence in Figure 4 (a). As shown, ELLA makes less mistakes than the baselines under various confidence thresholds. Figure 4 (b) displays how $K$ impacts the test NLL. We see that $K \in [20, 30]$ can already lead to good performance, reflecting the efficiency of ELLA. Another benefit of ELLA is that with it, we can actively control the performance vs. cost trade-off. Figure 4 (c) shows the comparison on the time used for predicting all CIFAR-10 test data. We note that ELLA is slightly slower than MFVI and substantially faster than *non-last-layer* LLA methods.

### 5.5 ImageNet Classification

We apply ELLA to ImageNet classification [8] to demonstrate its scalability. The experiment settings are identical with those for CIFAR-10. We observe that all LLA methods implemented with `Laplace` would cause out-of-memory (OOM) errors or suffer from very long fitting procedures, thus take only MAP and MFVI-BF as baselines. Table 2 presents the comparison on test accuracy, NLL, and ECE with ResNet architectures. We see that ELLA maintains its superiority in ECE while MFVI-BF can induce higher accuracy and lower NLL. This may be attributed to that the pretrained MAP model lies at a sharp maxima of the true posterior so it is necessary to properly adjust the mean of the Gaussian approximate posterior.

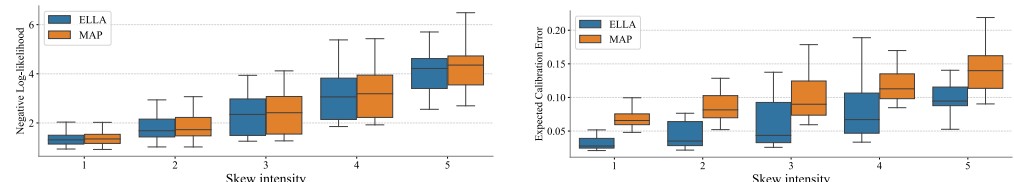

Figure 5: NLL (Left) and ECE (Right) on ImageNet corruptions for models trained with ViT-B architecture. Each box corresponds to a summary of the results across 19 types of skew.

We lastly apply ELLA to ViT-B [11]. We compare ELLA to MAP in Table 3 as all other baselines incur OOM errors or crushingly long running time. As shown, ELLA beats MAP in multiple aspects. Figure 5 shows the results of ELLA and MAP on ImageNet corruptions [19]. They are consistent with those for CIFAR-10 corruptions. We reveal by this experiment that ELLA can be a more applicable and scalable method than most of existing BNNs.

Table 3: Comparison on test accuracy (%) ↑, NLL ↓, and ECE ↓ on ImageNet with ViT-B architecture.

| Method | Acc. | NLL | ECE |
|--------|------|-------|-------|
| *ELLA* | 81.6 | 0.695 | 0.022 |
| *MAP*  | 81.5 | 0.700 | 0.039 |

## 6 Related Work

LA [38, 50, 14, 30, 22, 6, 7] locally approximates the Bayesian posterior with a Gaussian distribution, analogous to VI with Gaussian variationals [16, 1, 36, 52, 46] and SWAG [39]. LA can be applied to pretrained models effortlessly while the acquired posteriors are potentially restrictive (see Section 5.5). VI enjoys higher flexibility yet relies on costly training; BayesAdapter [10] seems to be a remedy to the issue but its accessibility is still lower than LA. SWAG stores a series of SGD iterates to heuristically construct an approximate posterior and is empirically weaker than LA/LLA [6].

Though more expressive approaches like deep ensembles [32] and MCMC [55, 4, 63] can explicitly explore diverse posterior modes, they face limitations in efficiency and scalability. What's more, it has been shown that LA/LLA can perform on par with or better than deep ensembles and cyclical MCMC on multiple benchmarks [6]. This may be attributed to the un-identified, complicated relationships between the parameters space and the function space of DNNs [59]. Also of note that LA can embrace deep ensembles to capture multiple posterior modes [13].

[9] introduces a general kernel approximation technique using neural networks. By contrast, we focus on leveraging kernel approximation to accelerate Laplace approximation. Thus, the focus of the two works is distinct. Indeed, this work shares a similar idea with the Sec 4.3 in [9] that LLA can be accelerated by kernel approximation. But, except for such an idea, this work differentiates from [9] in aspects like motivations, techniques, implementations, theoretical backgrounds, and applications. Our implementation, theoretical analysis, and some empirical findings are all novel.

## 7 Conclusion

This paper proposes ELLA, a simple, effective, and reliable approach for Bayesian deep learning. ELLA addresses the unreliability issues of existing approximations to LLA and is implemented based on Nyström method. We offer theoretical guarantees for ELLA and perform extensive studies to testify its efficacy and scalability. ELLA currently accounts for only the predictive, and extending it to estimate model evidence for model selection [38] deserves future investigation. Using model evidence to select the number of remaining eigenpairs $K$ for ELLA is also viable.

## Acknowledgments and Disclosure of Funding

This work was supported by the National Key Research and Development Program of China (No. 2017YFA0700904), NSF of China Projects (Nos. 62061136001, 62076145, 62106121, U19B2034, U1811461, U19A2081, 6197222); Beijing NSF Project (No. JQ19016); a grant from Tsinghua Institute for Guo Qiang; and the High Performance Computing Center, Tsinghua University. J.Z was also supported by the XPlorer Prize

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
