# A Proof

## A.1 Derivation of Equation (5)

Given

$$\mathbf{\Sigma} = \sigma_0^2 \Big( \mathbf{I}_P - \mathbf{J}_{\hat{\boldsymbol{\theta}},\mathbf{X}}^{\top} [\mathbf{\Lambda}_{\mathbf{X},\mathbf{Y}}^{-1}/\sigma_0^2 + \mathbf{J}_{\hat{\boldsymbol{\theta}},\mathbf{X}} \mathbf{J}_{\hat{\boldsymbol{\theta}},\mathbf{X}}^{\top}]^{-1} \mathbf{J}_{\hat{\boldsymbol{\theta}},\mathbf{X}} \Big),$$

we have

$$
\begin{aligned}
\kappa_{\mathrm{LLA}}(\boldsymbol{x}, \boldsymbol{x}') =& J_{\hat{\boldsymbol{\theta}}}(\boldsymbol{x}) \mathbf{\Sigma} J_{\hat{\boldsymbol{\theta}}}(\boldsymbol{x}')^{\top} \\
=& \sigma_0^2 J_{\hat{\boldsymbol{\theta}}}(\boldsymbol{x}) \left( \mathbf{I}_P - \mathbf{J}_{\hat{\boldsymbol{\theta}},\mathbf{X}}^{\top} \left[ \mathbf{\Lambda}_{\mathbf{X},\mathbf{Y}}^{-1}/\sigma_0^2 + \mathbf{J}_{\hat{\boldsymbol{\theta}},\mathbf{X}} \mathbf{J}_{\hat{\boldsymbol{\theta}},\mathbf{X}}^{\top} \right]^{-1} \mathbf{J}_{\hat{\boldsymbol{\theta}},\mathbf{X}} \right) J_{\hat{\boldsymbol{\theta}}}(\boldsymbol{x}')^{\top} \\
=& \sigma_0^2 \left( J_{\hat{\boldsymbol{\theta}}}(\boldsymbol{x}) J_{\hat{\boldsymbol{\theta}}}(\boldsymbol{x}')^{\top} - J_{\hat{\boldsymbol{\theta}}}(\boldsymbol{x}) \mathbf{J}_{\hat{\boldsymbol{\theta}},\mathbf{X}}^{\top} \left[ \mathbf{\Lambda}_{\mathbf{X},\mathbf{Y}}^{-1}/\sigma_0^2 + \mathbf{J}_{\hat{\boldsymbol{\theta}},\mathbf{X}} \mathbf{J}_{\hat{\boldsymbol{\theta}},\mathbf{X}}^{\top} \right]^{-1} \mathbf{J}_{\hat{\boldsymbol{\theta}},\mathbf{X}} J_{\hat{\boldsymbol{\theta}}}(\boldsymbol{x}')^{\top} \right) \\
=& \sigma_0^2 \Big( \kappa_{\mathrm{NTK}}(\boldsymbol{x}, \boldsymbol{x}') - \kappa_{\mathrm{NTK}}(\boldsymbol{x}, \mathbf{X}) [\mathbf{\Lambda}_{\mathbf{X},\mathbf{Y}}^{-1}/\sigma_0^2 + \kappa_{\mathrm{NTK}}(\mathbf{X}, \mathbf{X})]^{-1} \kappa_{\mathrm{NTK}}(\mathbf{X}, \boldsymbol{x}') \Big).
\end{aligned}
$$

## A.2 Derivation of Equation (6)

$$
\begin{aligned}
\kappa_{\mathrm{ELLA}}(\boldsymbol{x}, \boldsymbol{x}') =& \sigma_0^2 \left( \varphi(\boldsymbol{x}) \varphi(\boldsymbol{x}')^{\top} - \varphi(\boldsymbol{x}) \boldsymbol{\varphi}_{\mathbf{X}}^{\top} \left[ \mathbf{\Lambda}_{\mathbf{X},\mathbf{Y}}^{-1}/\sigma_0^2 + \boldsymbol{\varphi}_{\mathbf{X}} \boldsymbol{\varphi}_{\mathbf{X}}^{\top} \right]^{-1} \boldsymbol{\varphi}_{\mathbf{X}} \varphi(\boldsymbol{x}')^{\top} \right) \\
=& \varphi(\boldsymbol{x}) \sigma_0^2 \left( \mathbf{I}_K - \boldsymbol{\varphi}_{\mathbf{X}}^{\top} \left[ \mathbf{\Lambda}_{\mathbf{X},\mathbf{Y}}^{-1}/\sigma_0^2 + \boldsymbol{\varphi}_{\mathbf{X}} \boldsymbol{\varphi}_{\mathbf{X}}^{\top} \right]^{-1} \boldsymbol{\varphi}_{\mathbf{X}} \right) \varphi(\boldsymbol{x}')^{\top} \\
=& \varphi(\boldsymbol{x}) \left[ \boldsymbol{\varphi}_{\mathbf{X}}^{\top} \mathbf{\Lambda}_{\mathbf{X},\mathbf{Y}} \boldsymbol{\varphi}_{\mathbf{X}} + \mathbf{I}_K/\sigma_0^2 \right]^{-1} \varphi(\boldsymbol{x}')^{\top} \qquad \text{(Woodbury matrix identity)} \\
=& \varphi(\boldsymbol{x}) \left[ \sum_i \varphi(\boldsymbol{x}_i)^{\top} \Lambda(\boldsymbol{x}_i, \boldsymbol{y}_i) \varphi(\boldsymbol{x}_i) + \mathbf{I}_K/\sigma_0^2 \right]^{-1} \varphi(\boldsymbol{x}')^{\top}.
\end{aligned}
$$

## A.3 Derivation of Equation (18)

When $M = K$, the eigenvalues and eigenvectors of $\mathbf{K} = \mathbf{J}_{\hat{\boldsymbol{\theta}},\tilde{\mathbf{X}}} \mathbf{J}_{\hat{\boldsymbol{\theta}},\tilde{\mathbf{X}}}^{\top}$ can be organized as square matrices $\mathrm{diag}(\boldsymbol{\lambda}) = \mathrm{diag}(\lambda_1, ..., \lambda_K)$ and $U = [\boldsymbol{u}_1, ..., \boldsymbol{u}_K]$ respectively. And $\varphi(\boldsymbol{x}) = [J_{\hat{\boldsymbol{\theta}}}(\boldsymbol{x})\boldsymbol{v}_1, ..., J_{\hat{\boldsymbol{\theta}}}(\boldsymbol{x})\boldsymbol{v}_K] = J_{\hat{\boldsymbol{\theta}}}(\boldsymbol{x}) \mathbf{J}_{\hat{\boldsymbol{\theta}},\tilde{\mathbf{X}}}^{\top} U \mathrm{diag}(\boldsymbol{\lambda})^{-\frac{1}{2}}$. Of note that $U$ is a orthogonal matrix where $UU^{\top} = U^{\top}U = \mathbf{I}_K \Rightarrow U^{-1} = U^{\top}$, and by definition, $\mathbf{J}_{\hat{\boldsymbol{\theta}},\tilde{\mathbf{X}}} \mathbf{J}_{\hat{\boldsymbol{\theta}},\tilde{\mathbf{X}}}^{\top} = U \mathrm{diag}(\boldsymbol{\lambda}) U^{\top}$. Then

$$
\begin{aligned}
\kappa_{\mathrm{ELLA}}(\boldsymbol{x}, \boldsymbol{x}') =& \varphi(\boldsymbol{x}) \left[ \sum_i \varphi(\boldsymbol{x}_i)^{\top} \Lambda(\boldsymbol{x}_i, \boldsymbol{y}_i) \varphi(\boldsymbol{x}_i) + \mathbf{I}_K/\sigma_0^2 \right]^{-1} \varphi(\boldsymbol{x}')^{\top} \\
=& \varphi(\boldsymbol{x}) \left[ \boldsymbol{\varphi}_{\mathbf{X}}^{\top} \mathbf{\Lambda}_{\mathbf{X},\mathbf{Y}} \boldsymbol{\varphi}_{\mathbf{X}} + \mathbf{I}_K/\sigma_0^2 \right]^{-1} \varphi(\boldsymbol{x}')^{\top} \\
=& J_{\hat{\boldsymbol{\theta}}}(\boldsymbol{x}) \mathbf{J}_{\hat{\boldsymbol{\theta}},\tilde{\mathbf{X}}}^{\top} U \mathrm{diag}(\boldsymbol{\lambda})^{-\frac{1}{2}} \left[ \mathrm{diag}(\boldsymbol{\lambda})^{-\frac{1}{2}} U^{\top} \mathbf{J}_{\hat{\boldsymbol{\theta}},\tilde{\mathbf{X}}} \mathbf{J}_{\hat{\boldsymbol{\theta}},\mathbf{X}}^{\top} \mathbf{\Lambda}_{\mathbf{X},\mathbf{Y}} \mathbf{J}_{\hat{\boldsymbol{\theta}},\mathbf{X}} \mathbf{J}_{\hat{\boldsymbol{\theta}},\tilde{\mathbf{X}}}^{\top} U \mathrm{diag}(\boldsymbol{\lambda})^{-\frac{1}{2}} \right. \\
& \qquad\qquad\qquad\qquad \left. + \mathbf{I}_K/\sigma_0^2 \right]^{-1} \mathrm{diag}(\boldsymbol{\lambda})^{-\frac{1}{2}} U^{\top} \mathbf{J}_{\hat{\boldsymbol{\theta}},\tilde{\mathbf{X}}} J_{\hat{\boldsymbol{\theta}}}(\boldsymbol{x})^{\top} \\
=& J_{\hat{\boldsymbol{\theta}}}(\boldsymbol{x}) \mathbf{J}_{\hat{\boldsymbol{\theta}},\tilde{\mathbf{X}}}^{\top} U \left[ U^{\top} \mathbf{J}_{\hat{\boldsymbol{\theta}},\tilde{\mathbf{X}}} \mathbf{J}_{\hat{\boldsymbol{\theta}},\mathbf{X}}^{\top} \mathbf{\Lambda}_{\mathbf{X},\mathbf{Y}} \mathbf{J}_{\hat{\boldsymbol{\theta}},\mathbf{X}} \mathbf{J}_{\hat{\boldsymbol{\theta}},\tilde{\mathbf{X}}}^{\top} U + \mathrm{diag}(\boldsymbol{\lambda})/\sigma_0^2 \right]^{-1} U^{\top} \mathbf{J}_{\hat{\boldsymbol{\theta}},\tilde{\mathbf{X}}} J_{\hat{\boldsymbol{\theta}}}(\boldsymbol{x})^{\top} \\
=& J_{\hat{\boldsymbol{\theta}}}(\boldsymbol{x}) \mathbf{J}_{\hat{\boldsymbol{\theta}},\tilde{\mathbf{X}}}^{\top} \left[ UU^{\top} \mathbf{J}_{\hat{\boldsymbol{\theta}},\tilde{\mathbf{X}}} \mathbf{J}_{\hat{\boldsymbol{\theta}},\mathbf{X}}^{\top} \mathbf{\Lambda}_{\mathbf{X},\mathbf{Y}} \mathbf{J}_{\hat{\boldsymbol{\theta}},\mathbf{X}} \mathbf{J}_{\hat{\boldsymbol{\theta}},\tilde{\mathbf{X}}}^{\top} UU^{\top} + U \mathrm{diag}(\boldsymbol{\lambda}) U^{\top}/\sigma_0^2 \right]^{-1} \mathbf{J}_{\hat{\boldsymbol{\theta}},\tilde{\mathbf{X}}} J_{\hat{\boldsymbol{\theta}}}(\boldsymbol{x})^{\top} \\
=& J_{\hat{\boldsymbol{\theta}}}(\boldsymbol{x}) \mathbf{J}_{\hat{\boldsymbol{\theta}},\tilde{\mathbf{X}}}^{\top} \left[ \mathbf{J}_{\hat{\boldsymbol{\theta}},\tilde{\mathbf{X}}} \mathbf{J}_{\hat{\boldsymbol{\theta}},\mathbf{X}}^{\top} \mathbf{\Lambda}_{\mathbf{X},\mathbf{Y}} \mathbf{J}_{\hat{\boldsymbol{\theta}},\mathbf{X}} \mathbf{J}_{\hat{\boldsymbol{\theta}},\tilde{\mathbf{X}}}^{\top} + \mathbf{J}_{\hat{\boldsymbol{\theta}},\tilde{\mathbf{X}}} \mathbf{J}_{\hat{\boldsymbol{\theta}},\tilde{\mathbf{X}}}^{\top}/\sigma_0^2 \right]^{-1} \mathbf{J}_{\hat{\boldsymbol{\theta}},\tilde{\mathbf{X}}} J_{\hat{\boldsymbol{\theta}}}(\boldsymbol{x})^{\top}.
\end{aligned}
$$

## A.4 Proof of Theorem 1

**Theorem 1.** *Let $c_{\Lambda}$ be a finite constant associated with $\Lambda$, and $\mathcal{E}'$ the error of Nyström approximation $\|\mathbf{J}_{\hat{\boldsymbol{\theta}},\mathbf{X}} \mathbf{J}_{\hat{\boldsymbol{\theta}},\tilde{\mathbf{X}}}^{\top} (\mathbf{J}_{\hat{\boldsymbol{\theta}},\tilde{\mathbf{X}}} \mathbf{J}_{\hat{\boldsymbol{\theta}},\tilde{\mathbf{X}}}^{\top})^{-1} \mathbf{J}_{\hat{\boldsymbol{\theta}},\tilde{\mathbf{X}}} \mathbf{J}_{\hat{\boldsymbol{\theta}},\mathbf{X}}^{\top} - \mathbf{J}_{\hat{\boldsymbol{\theta}},\mathbf{X}} \mathbf{J}_{\hat{\boldsymbol{\theta}},\mathbf{X}}^{\top}\|$. It holds that*

$$\mathcal{E} \leq \sigma_0^4 c_{\Lambda} \mathcal{E}' + \sigma_0^2.$$

*Proof.* With $\mathbf{K} = \mathbf{J}_{\hat{\boldsymbol{\theta}},\tilde{\mathbf{X}}}\mathbf{J}_{\hat{\boldsymbol{\theta}},\tilde{\mathbf{X}}}^{\top} \succ 0$, let $\mathbf{P} = \mathbf{J}_{\hat{\boldsymbol{\theta}},\tilde{\mathbf{X}}}^{\top}\mathbf{K}^{-1}\mathbf{J}_{\hat{\boldsymbol{\theta}},\tilde{\mathbf{X}}}$. By Woodbury matrix identity,

$$
\begin{aligned}
\boldsymbol{\Sigma}' &= \mathbf{J}_{\hat{\boldsymbol{\theta}},\tilde{\mathbf{X}}}^{\top}\Big[\mathbf{J}_{\hat{\boldsymbol{\theta}},\tilde{\mathbf{X}}}\mathbf{J}_{\hat{\boldsymbol{\theta}},\mathbf{X}}^{\top}\boldsymbol{\Lambda}_{\mathbf{X},\mathbf{Y}}\mathbf{J}_{\hat{\boldsymbol{\theta}},\mathbf{X}}\mathbf{J}_{\hat{\boldsymbol{\theta}},\tilde{\mathbf{X}}}^{\top} + \mathbf{J}_{\hat{\boldsymbol{\theta}},\tilde{\mathbf{X}}}\mathbf{J}_{\hat{\boldsymbol{\theta}},\tilde{\mathbf{X}}}^{\top}/\sigma_0^2\Big]^{-1}\mathbf{J}_{\hat{\boldsymbol{\theta}},\tilde{\mathbf{X}}} \\
&= \mathbf{J}_{\hat{\boldsymbol{\theta}},\tilde{\mathbf{X}}}^{\top}\Big[\mathbf{J}_{\hat{\boldsymbol{\theta}},\tilde{\mathbf{X}}}\mathbf{J}_{\hat{\boldsymbol{\theta}},\mathbf{X}}^{\top}\boldsymbol{\Lambda}_{\mathbf{X},\mathbf{Y}}\mathbf{J}_{\hat{\boldsymbol{\theta}},\mathbf{X}}\mathbf{J}_{\hat{\boldsymbol{\theta}},\tilde{\mathbf{X}}}^{\top} + \mathbf{K}/\sigma_0^2\Big]^{-1}\mathbf{J}_{\hat{\boldsymbol{\theta}},\tilde{\mathbf{X}}} \\
&= \sigma_0^2\mathbf{J}_{\hat{\boldsymbol{\theta}},\tilde{\mathbf{X}}}^{\top}\Big[\mathbf{K}^{-1} - \mathbf{K}^{-1}\mathbf{J}_{\hat{\boldsymbol{\theta}},\tilde{\mathbf{X}}}\mathbf{J}_{\hat{\boldsymbol{\theta}},\mathbf{X}}^{\top}(\boldsymbol{\Lambda}_{\mathbf{X},\mathbf{Y}}^{-1}/\sigma_0^2 + \mathbf{J}_{\hat{\boldsymbol{\theta}},\mathbf{X}}\mathbf{J}_{\hat{\boldsymbol{\theta}},\tilde{\mathbf{X}}}^{\top}\mathbf{K}^{-1}\mathbf{J}_{\hat{\boldsymbol{\theta}},\tilde{\mathbf{X}}}\mathbf{J}_{\hat{\boldsymbol{\theta}},\mathbf{X}}^{\top})^{-1}\mathbf{J}_{\hat{\boldsymbol{\theta}},\mathbf{X}}\mathbf{J}_{\hat{\boldsymbol{\theta}},\tilde{\mathbf{X}}}^{\top}\mathbf{K}^{-1}\Big]\mathbf{J}_{\hat{\boldsymbol{\theta}},\tilde{\mathbf{X}}} \\
&= \sigma_0^2\Big[\mathbf{P} - \mathbf{P}\mathbf{J}_{\hat{\boldsymbol{\theta}},\mathbf{X}}^{\top}(\boldsymbol{\Lambda}_{\mathbf{X},\mathbf{Y}}^{-1}/\sigma_0^2 + \mathbf{J}_{\hat{\boldsymbol{\theta}},\mathbf{X}}\mathbf{P}\mathbf{J}_{\hat{\boldsymbol{\theta}},\mathbf{X}}^{\top})^{-1}\mathbf{J}_{\hat{\boldsymbol{\theta}},\mathbf{X}}\mathbf{P}\Big].
\end{aligned}
$$

It is interesting to see that $\mathbf{P}$ is a projector: $\mathbf{P}^2 = \mathbf{P}$ and $\mathbf{P}\mathbf{J}_{\hat{\boldsymbol{\theta}},\tilde{\mathbf{X}}}^{\top} = \mathbf{J}_{\hat{\boldsymbol{\theta}},\tilde{\mathbf{X}}}^{\top}$. Then

$$
\boldsymbol{\Sigma}' = \sigma_0^2\big[\mathbf{P} - \mathbf{P}\mathbf{J}_{\hat{\boldsymbol{\theta}},\mathbf{X}}^{\top}(\boldsymbol{\Lambda}_{\mathbf{X},\mathbf{Y}}^{-1}/\sigma_0^2 + \mathbf{J}_{\hat{\boldsymbol{\theta}},\mathbf{X}}\mathbf{P}\mathbf{P}\mathbf{J}_{\hat{\boldsymbol{\theta}},\mathbf{X}}^{\top})^{-1}\mathbf{J}_{\hat{\boldsymbol{\theta}},\mathbf{X}}\mathbf{P}\big].
$$

By Woodbury matrix identity again,

$$
\begin{aligned}
\boldsymbol{\Sigma}' = \sigma_0^2\Bigg[\mathbf{P} - \sigma_0^2\mathbf{P}\mathbf{J}_{\hat{\boldsymbol{\theta}},\mathbf{X}}^{\top}\Big(\boldsymbol{\Lambda}_{\mathbf{X},\mathbf{Y}} - \boldsymbol{\Lambda}_{\mathbf{X},\mathbf{Y}}\mathbf{J}_{\hat{\boldsymbol{\theta}},\mathbf{X}}\mathbf{P}(\mathbf{I}_P/\sigma_0^2 \\
+ \mathbf{P}\mathbf{J}_{\hat{\boldsymbol{\theta}},\mathbf{X}}^{\top}\boldsymbol{\Lambda}_{\mathbf{X},\mathbf{Y}}\mathbf{J}_{\hat{\boldsymbol{\theta}},\mathbf{X}}\mathbf{P})^{-1}\mathbf{P}\mathbf{J}_{\hat{\boldsymbol{\theta}},\mathbf{X}}^{\top}\boldsymbol{\Lambda}_{\mathbf{X},\mathbf{Y}}\Big)\mathbf{J}_{\hat{\boldsymbol{\theta}},\mathbf{X}}\mathbf{P}\Bigg] \\
= \sigma_0^2\Bigg[\mathbf{P} - \sigma_0^2\Big(\mathbf{P}\mathbf{J}_{\hat{\boldsymbol{\theta}},\mathbf{X}}^{\top}\boldsymbol{\Lambda}_{\mathbf{X},\mathbf{Y}}\mathbf{J}_{\hat{\boldsymbol{\theta}},\mathbf{X}}\mathbf{P} - \mathbf{P}\mathbf{J}_{\hat{\boldsymbol{\theta}},\mathbf{X}}^{\top}\boldsymbol{\Lambda}_{\mathbf{X},\mathbf{Y}}\mathbf{J}_{\hat{\boldsymbol{\theta}},\mathbf{X}}\mathbf{P}(\mathbf{I}_P/\sigma_0^2 \\
+ \mathbf{P}\mathbf{J}_{\hat{\boldsymbol{\theta}},\mathbf{X}}^{\top}\boldsymbol{\Lambda}_{\mathbf{X},\mathbf{Y}}\mathbf{J}_{\hat{\boldsymbol{\theta}},\mathbf{X}}\mathbf{P})^{-1}\mathbf{P}\mathbf{J}_{\hat{\boldsymbol{\theta}},\mathbf{X}}^{\top}\boldsymbol{\Lambda}_{\mathbf{X},\mathbf{Y}}\mathbf{J}_{\hat{\boldsymbol{\theta}},\mathbf{X}}\mathbf{P}\Big)\Bigg].
\end{aligned}
$$

Let $\mathbf{T} = (\mathbf{P}\mathbf{J}_{\hat{\boldsymbol{\theta}},\mathbf{X}}^{\top}\boldsymbol{\Lambda}_{\mathbf{X},\mathbf{Y}}\mathbf{J}_{\hat{\boldsymbol{\theta}},\mathbf{X}}\mathbf{P} + \mathbf{I}_P/\sigma_0^2)^{-1}$, so $\mathbf{P}\mathbf{J}_{\hat{\boldsymbol{\theta}},\mathbf{X}}^{\top}\boldsymbol{\Lambda}_{\mathbf{X},\mathbf{Y}}\mathbf{J}_{\hat{\boldsymbol{\theta}},\mathbf{X}}\mathbf{P} = \mathbf{T}^{-1} - \mathbf{I}_P/\sigma_0^2$. It follows that

$$
\boldsymbol{\Sigma}' = \sigma_0^2\Big[\mathbf{P} - \sigma_0^2\Big(\mathbf{T}^{-1} - \mathbf{I}_P/\sigma_0^2 - (\mathbf{T}^{-1} - \mathbf{I}_P/\sigma_0^2)\mathbf{T}(\mathbf{T}^{-1} - \mathbf{I}_P/\sigma_0^2)\Big)\Big] = \mathbf{T} + \sigma_0^2(\mathbf{P} - \mathbf{I}_P).
$$

As a result,

$$
\begin{aligned}
\mathcal{E} &= \|\boldsymbol{\Sigma}' - \boldsymbol{\Sigma}\| \\
&= \|\mathbf{T} + \sigma_0^2(\mathbf{P} - \mathbf{I}_P) - \boldsymbol{\Sigma}\| \\
&\leq \|\mathbf{T} - \boldsymbol{\Sigma}\| + \sigma_0^2\|\mathbf{P} - \mathbf{I}_P\| \\
&= \|\mathbf{T} - \boldsymbol{\Sigma}\| + \sigma_0^2,
\end{aligned}
$$

where we leverage the fact that the eigenvalue of the projector $\mathbf{P}$ is either $1$ or $0$. And

$$
\begin{aligned}
\|\mathbf{T} - \boldsymbol{\Sigma}\| &= \| - \mathbf{T}(\mathbf{T}^{-1} - \boldsymbol{\Sigma}^{-1})\boldsymbol{\Sigma}\| \\
&= \| - \mathbf{T}(\mathbf{P}\mathbf{J}_{\hat{\boldsymbol{\theta}},\mathbf{X}}^{\top}\boldsymbol{\Lambda}_{\mathbf{X},\mathbf{Y}}\mathbf{J}_{\hat{\boldsymbol{\theta}},\mathbf{X}}\mathbf{P} - \mathbf{J}_{\hat{\boldsymbol{\theta}},\mathbf{X}}^{\top}\boldsymbol{\Lambda}_{\mathbf{X},\mathbf{Y}}\mathbf{J}_{\hat{\boldsymbol{\theta}},\mathbf{X}})\boldsymbol{\Sigma}\| \\
&\leq \|\mathbf{T}\|\|\boldsymbol{\Sigma}\|\|\mathbf{P}\mathbf{J}_{\hat{\boldsymbol{\theta}},\mathbf{X}}^{\top}\boldsymbol{\Lambda}_{\mathbf{X},\mathbf{Y}}\mathbf{J}_{\hat{\boldsymbol{\theta}},\mathbf{X}}\mathbf{P} - \mathbf{J}_{\hat{\boldsymbol{\theta}},\mathbf{X}}^{\top}\boldsymbol{\Lambda}_{\mathbf{X},\mathbf{Y}}\mathbf{J}_{\hat{\boldsymbol{\theta}},\mathbf{X}}\| \\
&\leq \frac{\|\mathbf{P}\mathbf{J}_{\hat{\boldsymbol{\theta}},\mathbf{X}}^{\top}\boldsymbol{\Lambda}_{\mathbf{X},\mathbf{Y}}\mathbf{J}_{\hat{\boldsymbol{\theta}},\mathbf{X}}\mathbf{P} - \mathbf{J}_{\hat{\boldsymbol{\theta}},\mathbf{X}}^{\top}\boldsymbol{\Lambda}_{\mathbf{X},\mathbf{Y}}\mathbf{J}_{\hat{\boldsymbol{\theta}},\mathbf{X}}\|}{\lambda_{\min}(\mathbf{P}\mathbf{J}_{\hat{\boldsymbol{\theta}},\mathbf{X}}^{\top}\boldsymbol{\Lambda}_{\mathbf{X},\mathbf{Y}}\mathbf{J}_{\hat{\boldsymbol{\theta}},\mathbf{X}}\mathbf{P} + \mathbf{I}_P/\sigma_0^2)\lambda_{\min}(\mathbf{J}_{\hat{\boldsymbol{\theta}},\mathbf{X}}^{\top}\boldsymbol{\Lambda}_{\mathbf{X},\mathbf{Y}}\mathbf{J}_{\hat{\boldsymbol{\theta}},\mathbf{X}} + \mathbf{I}_P/\sigma_0^2)} \\
&\leq \sigma_0^4\|\mathbf{P}\mathbf{J}_{\hat{\boldsymbol{\theta}},\mathbf{X}}^{\top}\boldsymbol{\Lambda}_{\mathbf{X},\mathbf{Y}}\mathbf{J}_{\hat{\boldsymbol{\theta}},\mathbf{X}}\mathbf{P} - \mathbf{J}_{\hat{\boldsymbol{\theta}},\mathbf{X}}^{\top}\boldsymbol{\Lambda}_{\mathbf{X},\mathbf{Y}}\mathbf{J}_{\hat{\boldsymbol{\theta}},\mathbf{X}}\|,
\end{aligned}
$$

where $\lambda_{\min}(\cdot)$ denotes the smallest eigenvalue of a matrix. The last inequality holds due to that the eigenvalues of $\mathbf{P}\mathbf{J}_{\hat{\boldsymbol{\theta}},\mathbf{X}}^{\top}\boldsymbol{\Lambda}_{\mathbf{X},\mathbf{Y}}\mathbf{J}_{\hat{\boldsymbol{\theta}},\mathbf{X}}\mathbf{P} + \mathbf{I}_P/\sigma_0^2$ and $\mathbf{J}_{\hat{\boldsymbol{\theta}},\mathbf{X}}^{\top}\boldsymbol{\Lambda}_{\mathbf{X},\mathbf{Y}}\mathbf{J}_{\hat{\boldsymbol{\theta}},\mathbf{X}} + \mathbf{I}_P/\sigma_0^2$ are larger than or equal to $1/\sigma_0^2$ as $\mathbf{P}\mathbf{J}_{\hat{\boldsymbol{\theta}},\mathbf{X}}^{\top}\boldsymbol{\Lambda}_{\mathbf{X},\mathbf{Y}}\mathbf{J}_{\hat{\boldsymbol{\theta}},\mathbf{X}}\mathbf{P}$ and $\mathbf{J}_{\hat{\boldsymbol{\theta}},\mathbf{X}}^{\top}\boldsymbol{\Lambda}_{\mathbf{X},\mathbf{Y}}\mathbf{J}_{\hat{\boldsymbol{\theta}},\mathbf{X}}$ are SPSD.

To estimate the upper bound of $\|\mathbf{T} - \boldsymbol{\Sigma}\|$, we lay out the following Lemma.

**Lemma 1** (Proposition of Weinstein–Aronszajn identity). *If $\mathcal{A}$ and $\mathcal{B}$ are matrices of size $m \times n$ and $n \times m$ respectively, the non-zero eigenvalues of $\mathcal{A}\mathcal{B}$ and $\mathcal{B}\mathcal{A}$ are the same.*

It follows that

$$\|\mathbf{T}-\mathbf{\Sigma}\| \leq \sigma_0^4 \|\mathbf{P}\mathbf{J}_{\hat{\boldsymbol{\theta}},\mathbf{X}}^\top \mathbf{\Lambda}_{\mathbf{X},\mathbf{Y}}\mathbf{J}_{\hat{\boldsymbol{\theta}},\mathbf{X}}\mathbf{P} - \mathbf{J}_{\hat{\boldsymbol{\theta}},\mathbf{X}}^\top \mathbf{\Lambda}_{\mathbf{X},\mathbf{Y}}\mathbf{J}_{\hat{\boldsymbol{\theta}},\mathbf{X}}\|$$

$$=\frac{\sigma_0^4}{2}\|(\mathbf{P}\mathbf{J}_{\hat{\boldsymbol{\theta}},\mathbf{X}}^\top \mathbf{\Lambda}_{\mathbf{X},\mathbf{Y}}^{\frac{1}{2}} + \mathbf{J}_{\hat{\boldsymbol{\theta}},\mathbf{X}}^\top \mathbf{\Lambda}_{\mathbf{X},\mathbf{Y}}^{\frac{1}{2}})(\mathbf{\Lambda}_{\mathbf{X},\mathbf{Y}}^{\frac{1}{2}}\mathbf{J}_{\hat{\boldsymbol{\theta}},\mathbf{X}}\mathbf{P} - \mathbf{\Lambda}_{\mathbf{X},\mathbf{Y}}^{\frac{1}{2}}\mathbf{J}_{\hat{\boldsymbol{\theta}},\mathbf{X}})$$
$$+ (\mathbf{P}\mathbf{J}_{\hat{\boldsymbol{\theta}},\mathbf{X}}^\top \mathbf{\Lambda}_{\mathbf{X},\mathbf{Y}}^{\frac{1}{2}} - \mathbf{J}_{\hat{\boldsymbol{\theta}},\mathbf{X}}^\top \mathbf{\Lambda}_{\mathbf{X},\mathbf{Y}}^{\frac{1}{2}})(\mathbf{\Lambda}_{\mathbf{X},\mathbf{Y}}^{\frac{1}{2}}\mathbf{J}_{\hat{\boldsymbol{\theta}},\mathbf{X}}\mathbf{P} + \mathbf{\Lambda}_{\mathbf{X},\mathbf{Y}}^{\frac{1}{2}}\mathbf{J}_{\hat{\boldsymbol{\theta}},\mathbf{X}})\|$$

$$\leq\frac{\sigma_0^4}{2}\|(\mathbf{P}\mathbf{J}_{\hat{\boldsymbol{\theta}},\mathbf{X}}^\top \mathbf{\Lambda}_{\mathbf{X},\mathbf{Y}}^{\frac{1}{2}} + \mathbf{J}_{\hat{\boldsymbol{\theta}},\mathbf{X}}^\top \mathbf{\Lambda}_{\mathbf{X},\mathbf{Y}}^{\frac{1}{2}})(\mathbf{\Lambda}_{\mathbf{X},\mathbf{Y}}^{\frac{1}{2}}\mathbf{J}_{\hat{\boldsymbol{\theta}},\mathbf{X}}\mathbf{P} - \mathbf{\Lambda}_{\mathbf{X},\mathbf{Y}}^{\frac{1}{2}}\mathbf{J}_{\hat{\boldsymbol{\theta}},\mathbf{X}})\|$$
$$+ \frac{\sigma_0^4}{2}\|((\mathbf{P}\mathbf{J}_{\hat{\boldsymbol{\theta}},\mathbf{X}}^\top \mathbf{\Lambda}_{\mathbf{X},\mathbf{Y}}^{\frac{1}{2}} + \mathbf{J}_{\hat{\boldsymbol{\theta}},\mathbf{X}}^\top \mathbf{\Lambda}_{\mathbf{X},\mathbf{Y}}^{\frac{1}{2}})(\mathbf{\Lambda}_{\mathbf{X},\mathbf{Y}}^{\frac{1}{2}}\mathbf{J}_{\hat{\boldsymbol{\theta}},\mathbf{X}}\mathbf{P} - \mathbf{\Lambda}_{\mathbf{X},\mathbf{Y}}^{\frac{1}{2}}\mathbf{J}_{\hat{\boldsymbol{\theta}},\mathbf{X}}))^\top\|$$

$$=\sigma_0^4\|(\mathbf{P}\mathbf{J}_{\hat{\boldsymbol{\theta}},\mathbf{X}}^\top \mathbf{\Lambda}_{\mathbf{X},\mathbf{Y}}^{\frac{1}{2}} + \mathbf{J}_{\hat{\boldsymbol{\theta}},\mathbf{X}}^\top \mathbf{\Lambda}_{\mathbf{X},\mathbf{Y}}^{\frac{1}{2}})(\mathbf{\Lambda}_{\mathbf{X},\mathbf{Y}}^{\frac{1}{2}}\mathbf{J}_{\hat{\boldsymbol{\theta}},\mathbf{X}}\mathbf{P} - \mathbf{\Lambda}_{\mathbf{X},\mathbf{Y}}^{\frac{1}{2}}\mathbf{J}_{\hat{\boldsymbol{\theta}},\mathbf{X}})\| \quad (\|\mathbf{A}^\top\| = \|\mathbf{A}\|)$$

$$=\sigma_0^4\|(\mathbf{P} + \mathbf{I}_P)\mathbf{J}_{\hat{\boldsymbol{\theta}},\mathbf{X}}^\top \mathbf{\Lambda}_{\mathbf{X},\mathbf{Y}}\mathbf{J}_{\hat{\boldsymbol{\theta}},\mathbf{X}}(\mathbf{P} - \mathbf{I}_P)\|$$

$$=\sigma_0^4\|\mathbf{\Lambda}_{\mathbf{X},\mathbf{Y}}\mathbf{J}_{\hat{\boldsymbol{\theta}},\mathbf{X}}(\mathbf{P} - \mathbf{I}_P)(\mathbf{P} + \mathbf{I}_P)\mathbf{J}_{\hat{\boldsymbol{\theta}},\mathbf{X}}^\top\| \qquad \text{(Lemma 1)}$$

$$=\sigma_0^4\|\mathbf{\Lambda}_{\mathbf{X},\mathbf{Y}}\mathbf{J}_{\hat{\boldsymbol{\theta}},\mathbf{X}}(\mathbf{P} - \mathbf{I}_P)\mathbf{J}_{\hat{\boldsymbol{\theta}},\mathbf{X}}^\top\| \qquad (\mathbf{P}^2 = \mathbf{P})$$

$$=\sigma_0^4\|\mathbf{\Lambda}_{\mathbf{X},\mathbf{Y}}(\mathbf{J}_{\hat{\boldsymbol{\theta}},\mathbf{X}}\mathbf{J}_{\hat{\boldsymbol{\theta}},\tilde{\mathbf{X}}}^\top (\mathbf{J}_{\hat{\boldsymbol{\theta}},\tilde{\mathbf{X}}}\mathbf{J}_{\hat{\boldsymbol{\theta}},\tilde{\mathbf{X}}}^\top)^{-1}\mathbf{J}_{\hat{\boldsymbol{\theta}},\tilde{\mathbf{X}}}\mathbf{J}_{\hat{\boldsymbol{\theta}},\mathbf{X}}^\top - \mathbf{J}_{\hat{\boldsymbol{\theta}},\mathbf{X}}\mathbf{J}_{\hat{\boldsymbol{\theta}},\mathbf{X}}^\top)\|$$

$$\leq\sigma_0^4\|\mathbf{\Lambda}_{\mathbf{X},\mathbf{Y}}\|\|\mathbf{J}_{\hat{\boldsymbol{\theta}},\mathbf{X}}\mathbf{J}_{\hat{\boldsymbol{\theta}},\tilde{\mathbf{X}}}^\top (\mathbf{J}_{\hat{\boldsymbol{\theta}},\tilde{\mathbf{X}}}\mathbf{J}_{\hat{\boldsymbol{\theta}},\tilde{\mathbf{X}}}^\top)^{-1}\mathbf{J}_{\hat{\boldsymbol{\theta}},\tilde{\mathbf{X}}}\mathbf{J}_{\hat{\boldsymbol{\theta}},\mathbf{X}}^\top - \mathbf{J}_{\hat{\boldsymbol{\theta}},\mathbf{X}}\mathbf{J}_{\hat{\boldsymbol{\theta}},\mathbf{X}}^\top\|.$$

It is easy to show that $\|\mathbf{\Lambda}_{\mathbf{X},\mathbf{Y}}\| \leq c_\Lambda$ with $c_\Lambda$ as a finite constant. For example, when handling regression tasks, $-\log p(\boldsymbol{y}|\boldsymbol{g})$ typically boils down to $\frac{1}{2\sigma_{\text{noise}}^2}(\boldsymbol{g} - \boldsymbol{y})^\top(\boldsymbol{g} - \boldsymbol{y})$ where $\sigma_{\text{noise}}^2$ refers to the variance of data noise, so $-\nabla_{\boldsymbol{gg}}^2 \log p(\boldsymbol{y}|\boldsymbol{g}) = \frac{1}{\sigma_{\text{noise}}^2}\mathbf{I}_C$ and $c_\Lambda = \frac{1}{\sigma_{\text{noise}}^2}$.

When facing classification cases, $-\log p(\boldsymbol{y}|\boldsymbol{g})$ becomes the softmax cross-entropy loss. It is guaranteed that $c_\Lambda \leq 2$ because

$$\| - \nabla_{\boldsymbol{gg}}^2 \log p(\boldsymbol{y}|\boldsymbol{g})\| = \|\operatorname{diag}(\operatorname{softmax}(\boldsymbol{g})) - \operatorname{softmax}(\boldsymbol{g})\operatorname{softmax}(\boldsymbol{g})^\top\|$$
$$\leq \|\operatorname{diag}(\operatorname{softmax}(\boldsymbol{g}))\| + \|\operatorname{softmax}(\boldsymbol{g})\operatorname{softmax}(\boldsymbol{g})^\top\|$$
$$\leq 1 + \|\operatorname{softmax}(\boldsymbol{g})^\top \operatorname{softmax}(\boldsymbol{g})\| \leq 1 + 1 = 2.$$

To summarize, we have the following bound for the approximation error $\mathcal{E}$:

$$\mathcal{E} \leq \|\mathbf{T}-\mathbf{\Sigma}\| + \sigma_0^2 = \sigma_0^4 c_\Lambda\|\mathbf{J}_{\hat{\boldsymbol{\theta}},\mathbf{X}}\mathbf{J}_{\hat{\boldsymbol{\theta}},\tilde{\mathbf{X}}}^\top (\mathbf{J}_{\hat{\boldsymbol{\theta}},\tilde{\mathbf{X}}}\mathbf{J}_{\hat{\boldsymbol{\theta}},\tilde{\mathbf{X}}}^\top)^{-1}\mathbf{J}_{\hat{\boldsymbol{\theta}},\tilde{\mathbf{X}}}\mathbf{J}_{\hat{\boldsymbol{\theta}},\mathbf{X}}^\top - \mathbf{J}_{\hat{\boldsymbol{\theta}},\mathbf{X}}\mathbf{J}_{\hat{\boldsymbol{\theta}},\mathbf{X}}^\top\| + \sigma_0^2.$$

$\square$

# B    How Does ELLA Address the Scalability Issues of Nyström Approximation

Existing works like NeuralEF [9] highlight the scalability issues of the Nyström approximation when applied to NTKs, but they have been successfully addressed by ELLA, which reflects our technical novelty. On the one hand, the cost of eigendecomposing the $M \times M$ matrix in our case is low considering that we set $M < 10^4$. As shown in Figure 2(a)(b), $M = 2000$ can lead to reasonably small approximation errors for both the Nyström approximation and the resulting ELLA covariance. On the other hand, the prediction cost at test time is reduced by two innovations in this work. Firstly, looking at Equation (14), typical Nyström method first computes $J_{\hat{\boldsymbol{\theta}}}(\boldsymbol{x},i)\mathbf{J}_{\hat{\boldsymbol{\theta}},\tilde{\mathbf{X}}}^\top$ (i.e., evaluate the NTK similarities between $(\boldsymbol{x},i)$ and all training data $\tilde{\mathbf{X}}$) and then multiplies the result with $\mathbf{u}_k \in \mathbb{R}^M$ to get the evaluation of $k$-th eigenfunction. So it needs to store $\mathbf{J}_{\hat{\boldsymbol{\theta}},\tilde{\mathbf{X}}} \in \mathbb{R}^{M \times P}$ and compute $M$ Jacobian-vector products (JVPs). But this work proposes to first compute $\mathbf{v}_k = \mathbf{J}_{\hat{\boldsymbol{\theta}},\tilde{\mathbf{X}}}^\top \mathbf{u}_k, k = 1, ..., K$ (we omit the scalar multiplier) and then obtain the evaluation of $k$-th eigenfunction by $J_{\hat{\boldsymbol{\theta}}}(\boldsymbol{x},i)\mathbf{v}_k$. Namely, we only need to store $K$ vectors of size $P$ and compute $K$ JVPs. In most cases, $M = 2000$ and $K = 20$, so the benefits of our technique are obvious. Secondly, we leverage forward-mode autodiff (fwAD) to efficiently compute the JVPs where we do not need to explicitly compute and

Table 4: Test NLL of ELLA varies w.r.t. $M$ and $K$ ($M \geq K$) on CIFAR-10 with ResNet-20 architecture. The experiments for $K \geq 256$ are time-consuming so we have not included the corresponding results here.

| $M \backslash K$ | 4 | 8 | 16 | 32 | 64 | 128 |
|---|---|---|---|---|---|---|
| 4 | 0.2689 | | | | | |
| 8 | 0.2690 | 0.2561 | | | | |
| 16 | 0.2660 | 0.2548 | 0.2392 | | | |
| 32 | 0.2655 | 0.2541 | 0.2395 | 0.2318 | | |
| 64 | 0.2672 | 0.2527 | 0.2398 | 0.2312 | 0.2299 | |
| 128 | 0.2674 | 0.2540 | 0.2383 | 0.2310 | 0.2298 | 0.2294 |
| 256 | 0.2679 | 0.2545 | 0.2382 | 0.2310 | 0.2300 | 0.2294 |
| 512 | 0.2678 | 0.2542 | 0.2376 | 0.2315 | 0.2292 | 0.2288 |
| 1024 | 0.2674 | 0.2545 | 0.2385 | 0.2314 | 0.2295 | 0.2292 |
| 2000 | 0.2673 | 0.2551 | 0.2380 | 0.2308 | 0.2301 | 0.2295 |

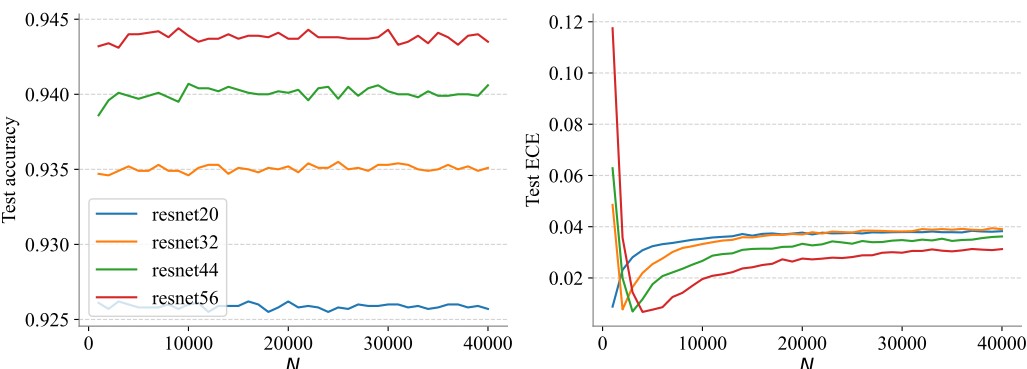

Figure 6: The test accuracy and ECE of ELLA vary w.r.t. the number of training data $N$ on CIFAR-10.

store the Jacobian $J_{\hat{\theta}}(\boldsymbol{x}, i)$ but to invoke once fwAD. Further, fwAD enables the parallel evaluation of JVPs over all output dimensions, i.e., we can concurrently compute $J_{\hat{\theta}}(\boldsymbol{x}, i)\mathbf{v}_k, i = 1, ..., C$ in one forward pass. These techniques make the evaluation of ELLA equal to performing $K$ forward passes under the scope of fwAD. This is similar to other BNNs that perform $S$ forward passes with different MC parameter samples to estimate the posterior predictive.

## C  More Experimental Results

### C.1  Test NLL of ELLA Varies w.r.t. K and M

Table 4 presents how the test NLL of ELLA varies w.r.t. $M$ and $K$ on CIFAR-10 with ResNet-20 architecture. Considering the results in Figure 2, we conclude that under the same $K$, a larger $M$ can bring closer approximation but it does not necessarily lead to a better test NLL.

### C.2  Test Accuracy and ECE of ELLA Vary w.r.t. N

Figure 6 presents how the test accuracy and ECE of ELLA vary w.r.t. the number of training data $N$ on CIFAR-10.

### C.3  Test NLL of LLA* Varies w.r.t. N

Figure 7 shows how the test NLL of LLA$^*$ varies w.r.t. the number of training data $N$ on CIFAR-10. These results, as well as those in Section 5.2, reflect that the overfitting issue of LLA generally exists. We can also see that properly tuning the prior precision can alleviate the overfitting of LLA$^*$ to some extent but cannot fully resolve it.

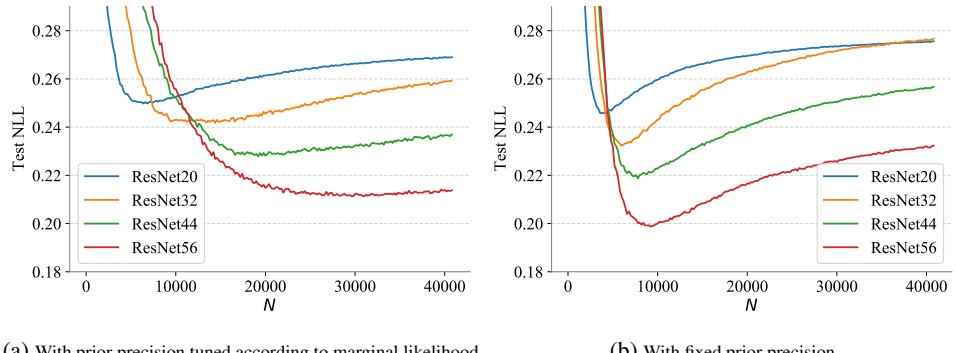

(a) With prior precision tuned according to marginal likelihood

(b) With fixed prior precision

Figure 7: The test NLL of LLA* varies w.r.t. the number of training data $N$ on CIFAR-10. The y axis is aligned with that of Figure 2(c).

Table 5: Comparison on the approximation error to vanilla LLA in the illustrative regression case, measured by the KL divergence between the predictive distributions.

|  | ELLA | LLA-KFAC | LLA-Diag | LLA* |
|---|---|---|---|---|
| KL div. | 0.83 | 0.35 | 1.71 | 2.44 |

### C.4    Comparison on the Approximation Error to Vanilla LLA

Table 5 lists the discrepancies between the predictive distributions of the approximate LLA method and vanilla LLA in the illustrative regression case detailed in Section 5.3. Here, considering the predictive distribution for one test datum is a Gaussian distribution, we use the KL divergence between the Gaussians yielded by the method of concern and LLA as a proxy of the approximation error (averaged over a set of test points).

### C.5    More Results on CIFAR-10 Corruptions

Figure 8, Figure 9, and Figure 10 show the results of the considered methods on CIFAR-10 corruptions using ResNet-20, ResNet-32, and ResNet-44 architectures respectively. We can see that ELLA surpasses the baselines in aspects of NLL and ECE at various levels of skew. These results signify ELLA's ability to make conservative predictions for OOD inputs.

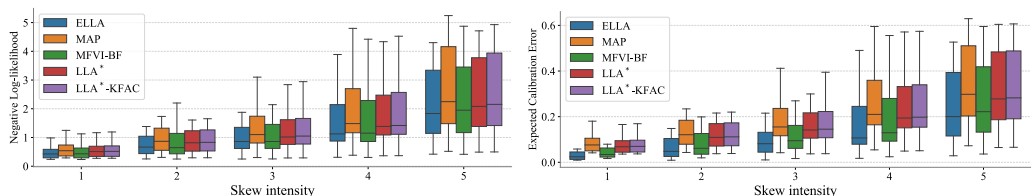

Figure 8: NLL (Left) and ECE (Right) on CIFAR-10 corruptions for models trained with ResNet-20 architecture. Each box corresponds to a summary of the results across 19 types of skew.

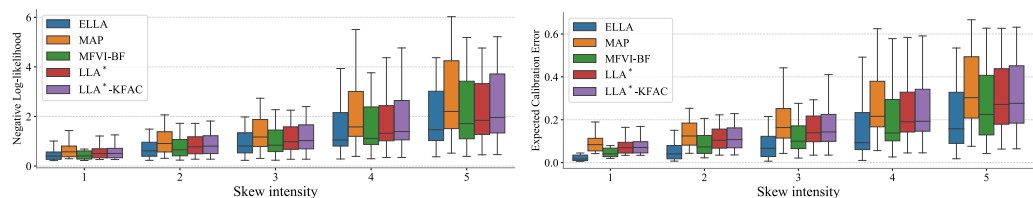

Figure 9: NLL (Left) and ECE (Right) on CIFAR-10 corruptions for models trained with ResNet-32 architecture. Each box corresponds to a summary of the results across 19 types of skew.

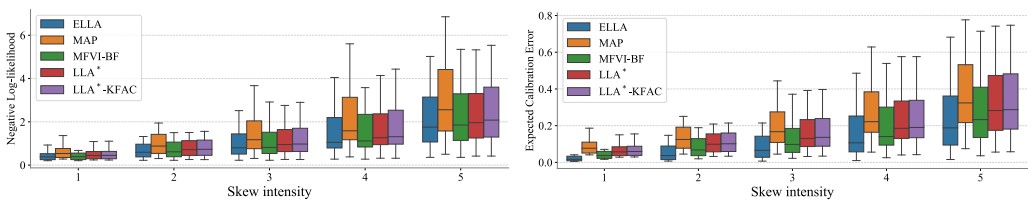

Figure 10: NLL (Left) and ECE (Right) on CIFAR-10 corruptions for models trained with ResNet-44 architecture. Each box corresponds to a summary of the results across 19 types of skew.