# OpenReview forum: "Accelerated Linearized Laplace Approximation for Bayesian Deep Learning"
_NeurIPS.cc/2022/Conference — NeurIPS 2022 Accept_

### Official Review · Reviewer_AKXi · 2022-07-06

**Rating:** 5
**Confidence:** 4
**Soundness:** 3 good
**Presentation:** 3 good
**Contribution:** 2 fair

**Summary:**

The paper targets the improvement of probabilistic predictions in Bayesian deep learning with the (linearized) Laplace approximation. In particular, it proposes to apply the Nystrom kernel approximation to the neural-tangent-like covariance matrix of the corresponding Gaussian process that is the function-space dual of the linearized Laplace in weight-space. Using the Nystrom approximation allows the authors to scale the linearized Laplace to ImageNet and ViT scale. They compare their method to other Laplace approximations in weight space and find that their method performs better, though in some cases only marginally, at a similar or lower computational cost.


**Questions:**

* I found it hard to understand/follow the discussion of the overfitting issue/behaviour of ELLA in Sec 5.2. As far as I understood, ELLA uses pre-trained models and only deals with the GP predictive distribution. As such, "early stopping" does not really make sense to me as it typically refers to early stopping of training. Could you please elaborate?

**Limitations:**

* see Weaknesses above, in particular, the authors only investigate ELLA for predictions.
* several works (e.g. ref [22], [6]) point out that the choice of prior precision is important to achieve good predictive performance (in particular for uncertainties/NLL/ECE); however, it looks like the values here have simply been taken from the pre-trained model regularisation, which may explain the poor performance of LLA and LLA+KFAC. For this reason, [6] (as far as I understand) proposes to use last-layer Laplace by default when these hyperparameters are not tuned/the MAP model itself does not overfit.
* the illustrative regression example (sec 5.3) is not discussed in enough detail. In particular, it should be investigated and further discussed why ELLA can be better than the full GGN model (LLA), to which it should be equivalent (except for the Nystrom approximation, of course). The authors are "surprised" (see caption to Fig 1) by ELLA performing better than the full model (LLA) as well, yet they do not investigate or discuss this. For example, the full GP predictive (without Nystrom approximation) as well as different/larger values for M and K should be considered in this case to build further intuition for the model and to investigate this behaviour.

**Strengths And Weaknesses:**

Strengths:
* The Laplace approximation for larger scale machine learning has recently seen increasing attention and this paper further improves scalability by introducing the Nystrom approximation in its dual (function space) formulation.
* The paper seems to be technically correct, though I haven't checked all the details.

Weaknesses:
* Generality + Novelty: The novelty and originality presented in this paper is limited, especially when compared to other papers at ICML or NeurIPS in this space (e.g. ref [6] provides a library for Laplace approximation and a thorough review/benchmark, or ref [9] introduces kernel approximations more generally and not only for large-scale Laplace predictions). While this paper extends the Laplace approximation to ImageNet and ViTs, the paper ultimately discusses one particular approximation (Nystrom) to one particular covariance function (NTK) for one particular task (probabilistic predictions on pre-trained models); it largely builds on existing libraries and improvements over MAP predictions are often marginal (especially on some of the larger-scale problems) and provided without errorbars. The authors also claim that no other methods can be applied to ViT or ImageNet, yet last-layer Laplace (LLA*) should be applicable in this case as well (at least when subsampling the dataset; e.g. [6] apply last layer Laplace to transformer models).
* Contributions + relation to prior work:
  - Several results in Sec 3.2 and 3.3 are, in my opinion, not novel and have been similarly derived in the kernel literature (where the Nystrom approximation is commonplace) as well as in Appendix A.2 and A.3 of NeuralEF (ICML2022, reference [9] in the paper under review) for the NTK kernel. While "NeuralEF" is referenced (as ref [9]) in the paper, it is not mentioned in Sec 3.2 and 3.3 of this paper and it is not clarified, which results in Sec 3.2 and 3.3 are novel and which are already known.
  - The Nystrom approximation is a well known and well studied approximation to kernel functions, yet a discussion/presentation of this is missing from this paper.
  - NeuralEF [9] highlights several limitations of the Nystrom approximation when applied to NTK mostly to do with scalability (see Sec 2.2 in [9] in particular). They highlight the cost of the eigendecomposition as well as prediction costs at test time. I agree with [9] that this could be problems and scalability of the Nystrom should be discussed more prominently.
* also see "Limitations" below
* I found the presentation/clarity to be not clear:
	- The setting (probabilistic prediction with Bayesian neural networks), ultimate goal/objective (improve linearized(?) posterior predictive), as well as the baseline methods (discussion + equations for e.g. Laplace with KFAC, ...) should be introduced more clearly. E.g. the posterior predictive distributions  evaluated are only mentioned in passing/do not have clear equations/equation numbers
	- The paper ultimately presents a technical contribution (Nystrom for this particular kernel) in an arguably very technical way. In my opinion, more emphasis should be put on explanation and some of the technical details (e.g. indexing with $i$ in Sec 3.3) should be left to the appendix.
	- Alg 1 and 2 should be explained in much more detail; comments + references to the text as well as from the text to the algorithms are missing

---

> ### Author Response · Authors · 2022-08-01
> **Response (part III/III)**
>
>
> ### Q10: Unclear settings and details
>
> Thanks for pointing out this. The setting is to make probabilistic predictions with Bayesian neural networks (see L63-71). The goal is to accelerate linearized Laplace with reassuring theoretical guarantees. The baselines like LA and LLA are detailed in Section 2.1 (though we have not exposited the formulations of KFAC/Diagonal approximations; we will fix this).
>
> We clarify that the posterior predictive has been detailed in both the parameter-space view (L71) and the function-space view (L205).
>
>
> ### Q11: More emphasis should be put on the explanation
>
> We appreciate the suggestion and will adjust the dense technical details. We clarify that the indexing $i$ in Sec 3.3 is defined in L131: it is a scalar indicating the Jacobian corresponding to the $i$-th output neuron of an NN.
>
>
> ### Q12: Alg 1 and 2 should be explained in much more detail
>
> We will fix the issues. However, we emphasize that after reading Sec 3.3 and 3.4, it should be easy to follow Alg 1 and 2 (e.g., Reviewer r6Sm states "I appreciate the implementation-specific details in Section 3.4 including the code snippets in Algorithm 1-2").
>
>
> ### Q13: Regarding over-fitting and early stopping
>
> The overfitting issue refers to that the test loss of ELLA (and LLA methods more generally) starts increasing after a certain number of data points having been used for fitting the covariance matrix. Then early stopping targets the fitting process of the covariance matrix (see L216-218), and it has nothing to do with the pre-trained models. Another benefit of early stopping is that it speeds up the fitting process. We will make this clearer in the revision. We also emphasize that the other reviewers like the experiment showing that ELLA tends to "overfit" to larger datasets and find it interesting.
>
> ### Q14: The authors only investigate ELLA for predictions
>
> We have pointed out that extending ELLA to estimate model evidence for model selection is possible in the conclusion section (see L284). Due to limited energy, we are currently focusing on the *typical* task -- improving the posterior predictive of Laplace approximation (previous works like [50, 22] also focus on this). The contributions of this paper on this point are unignorable and are appreciated by Reviewer 2eHY and Reviewer r6Sm. Considering the tight rebuttal period, we will try to include initial results on using model evidence for model selection in the revision (because the experimental settings for model selection are significantly different from those in the current manuscript).
>
>
> ### Q15: Regarding the choice of prior precision
>
> We are sorry that we have not clarified the experimental details for the Laplace baselines in the manuscript. In fact, we set the prior precision according to pre-trained model regularization *only* for ELLA. For all Laplace baselines (both full-layer and last-layer), we enable the automatic tuning of the prior precision in all experiments by invoking the `la.optimize_prior_precision` method (provided by the Laplace library; see the submitted code). Thus, the comparison between ELLA and the Laplace baselines should be fair, and the poor performance of LLA and LLA+KFAC reflects the superiority of ELLA.
>
>
> ### Q16: Regarding the illustrative regression example
>
> Thanks for the comments. We will revise the paper. We first point out that LLA is likely to underestimate the in-between uncertainty in this setting (we test an NN-GP model on this problem and observe higher in-between uncertainty). As detailed in Sec 5.3, ELLA relies on the Nystrom approximation of a $16\times 16$ matrix ($N=M=16$) with rank $5$ ($K=5$), so it should not be equivalent to the original LLA (i.e., the full GP predictive without Nystrom approximation). Such inequivalence incurs better in-between uncertainty for ELLA, and the underlying reason deserves further investigation. We will also test different values for $K$ to check the generality of this behavior.

---

> ### Author Response · Authors · 2022-08-01
> **Response (part II/III)**
>
> ### Q7: Regarding the ImageNet experiments
>
> We launch a new experiment using LLA*-KFAC with ResNet-18 architecture on ImageNet. LLA\*-KFAC is arguably more flexible and expressive than LLA\*-Diag. The hardware is an NVIDIA A100 GPU with 40G memory. The results as well as those of ELLA are presented in the following table.
>
>
> |   |      Acc.(%)      |  NLL |  ECE  |
> |----------|:-------------:|:-------------:|:-------------:|
> | ELLA |  **69.8** | **1.243** | 0.016 |
> | LLA*-FKAC |    69.6   |   1.246 | **0.015** |
> Table: Comparison on test accuracy (%), NLL, and ECE on ImageNet with ResNet-18 architecture.
>
> As shown, ELLA demonstrates better accuracy and NLL but slightly worse ECE than LLA\*-KFAC. This may be attributed to that we explicitly tune the prior precision for LLA*-KFAC but not for ELLA (in all experiments; see the call of `la.optimize_prior_precision` in the code). Considering LLA\*-KFAC requires higher memory consumption (at least in the implementation of the Laplace library; see the Issue panel of its GitHub repo) and longer fitting (as ELLA easily embraces early stopping), ELLA is still a better choice in ImageNet settings. We will add the ImageNet results with more architectures in the revision.
>
>
> ### Q8: Discussion on Nystrom approximation
>
> Thanks for the advice. In fact, we have done this in L126-129, and we also review some details of Nystrom method in the derivation of the approximation to NTKs in Section 3.3. Anyway, we will discuss more related works regarding Nystrom approximation in the revision.
>
>
> ### Q9: NeuralEF [9] highlights several limitations of the Nystrom approximation when applied to NTK mostly to do with scalability
>
> Yes, the limitations highlighted by [9] exist but have been successfully addressed by this paper, which reflects our technical novelty. On the one hand, the cost of eigendecomposing the $M \times M$ matrix in our case is low considering that we set $M < 10^4$. As shown in Figure 2(a)(b), $M=2000$ can lead to reasonably small approximation errors for both the Nystrom approximation and the resulting ELLA covariance. On the other hand, the prediction cost at test time is reduced by two innovations in this paper. Firstly, looking at Eq (14), typical Nystrom method first computes $J_{\hat{\theta}}(x, i) J_{\hat{\theta},\tilde{X}}^\top$ (i.e., evaluate the NTK kernel similarities between $(x, i)$ and all training data $\tilde{X}$) and then multiplies the result with $u_k \in \mathbb{R}^M$ to get the evaluation of $k$-th eigenfunction. So it needs to store $J_{\hat{\theta},\tilde{X}}\in \mathbb{R}^{M\times P}$ and compute $M$ Jacobian-vector products (JVPs). But this work proposes to first compute $v_k=J_{\hat{\theta},\tilde{X}}^\top u_k, k=1,...,K$ (we omit the scalar multiplier) and then obtain the evaluation of $k$-th eigenfunction by $J_{\hat{\theta}}(x, i) v_k$. Namely, we only need to store $K$ vectors of size $P$ and compute $K$ JVPs. In most cases, $M=2000$ and $K=20$, so the benefits of our technique are obvious. Secondly, we leverage forward-mode autodiff (fwAD) to efficiently compute the JVPs where we do not need to explicitly compute and store the Jacobian $J_{\hat{\theta}}(x, i)$ but to invoke once fwAD. Further, fwAD enables the parallel evaluation of JVPs over all the output dimensions, i.e., we can concurrently compute $J_{\hat{\theta}}(x,i) v_k, i=1,...,C$ in one forward pass. These techniques make the evaluation of ELLA equal to performing $K$ forward passes under the scope of fwAD. This is similar to other BNNs that perform $S$ forward passes with different MC parameter samples to estimate the posterior predictive.

---

> ### Author Response · Authors · 2022-08-01
> **Response (part I/III)**
>
> We thank the reviewer for the constructive comments. We make the following clarification.
>
> ### Q1: Comparison to [6]
>
> As pointed out by the reviewer, [6] provides a library for Laplace approximation and a thorough review/benchmark. By contrast, this paper strives to accelerate Laplace approximation to make it easily scale up to large data and deep architectures. In detail, the paper provides a new algorithm based on Nyström approximation, an efficient implementation with fwAD, insightful theoretical analysis, as well as extensive empirical results. That said, the two works make significantly different contributions to the community.
>
> ### Q2: Comparison and relation to [9]
>
> [9] introduces a general kernel approximation technique using neural networks, while this paper focuses on leveraging kernel approximation to accelerate Laplace approximation. As said, the focus of the two papers is distinct. Indeed, this paper shares a similar idea with the Sec 4.3 of [9] that LLA can be accelerated by kernel approximation. But, honestly, this is a very straightforward idea, which is explicitly pointed out by all three reviewers. Except for such an idea, this paper differentiates from [9] in aspects like motivations, techniques (with Nystrom approximation), implementations (with fwAD), theoretical backgrounds (bound of approximation errors), and applications. Our implementation, theoretical analysis, and some empirical findings are all novel. We will add a direct discussion on these points in the revision. We will also clarify in Sec 3.2 that Eq (6) has been similarly derived by [9], and clarify that the results in the following sections are newly derived.
>
> ### Q3: The paper ultimately discusses one particular approximation (Nystrom) to one particular covariance function (NTK) for one particular task
>
> We politely disagree that this is a limitation of this work. Recall that the acceptance standard (rating 7) for a NeurIPS paper is "technically solid paper, with high impact on at least one sub-area, or moderate-to-high impact on more than one areas". So what we have done – focusing on developing techniques to solve a specific problem – can be a reasonable contribution to the community. Of note that pioneering works like [50, 22] are in this line as well. We also highlight the reviews given by the other reviewers here: "overall convinces that the proposed method is a valuable addition to the family of Laplace approximations", "the proposed ELLA method for accelerating LLA is novel, theoretically sound, and intuitively sensible; it is therefore a timely and welcome addition to the fast-growing Laplace toolbox for Bayesian deep learning", "I particularly appreciate the inclusion of experiments on ImageNet, which is (unfortunately) still not so common in the Bayesian deep learning literature; I was especially impressed by the results on the recent Vision Transformer architecture".
>
> ### Q4: It largely builds on existing libraries
>
> This is a misunderstanding. As you can see from the algorithms in the paper and the submitted codebase, ELLA has a significantly different implementation from the typical Laplace approximations in the Laplace library. As a result, ELLA enjoys higher efficiency than typical LLA-KFAC and LLA-Diag implemented with the Laplace library (see Figure 4c), and is more flexible (e.g., compatible with early stopping in the fitting phase).
>
> ### Q5: Marginal improvements over MAP predictions
>
> In the aspect of accuracy, yes, the improvements over MAP are marginal, and this may be an internal issue of Laplace approximation methods. As for NLL, ECE, and OOD generalization/robustness, we respectfully disagree that the improvements are marginal (see Figure 3&4&5).
>
> ### Q6: Regarding error bars
>
> Thanks. Following the suggestion, we re-run the CIFAR-10 and ImageNet experiments with random seeds and have updated Table 1 and Table 2 (see the new results in [the reply to the first question of Reviewer r6Sm](https://openreview.net/forum?id=jftNpltMgz&noteId=AO6d4uNbqZx)).
>
> We hope these results can provide convincing statistical evidence for the superiority of the proposed method. Feel free to comment if there are further questions.

---

> > ### Comment · Reviewer_AKXi · 2022-08-04
> > **Thank you for your clarifications**
> >
> > I thank the authors for their thoughtful and detailed rebuttal as well as the clarifications to my questions. I appreciate it and will adapt my score to some degree to reflect this (after the rebuttal/discussion with the other reviewers). Here I clarify some of my comments and suggestions (no further response required). I'll post a few follow-up questions in a separate post.
> >
> > ## Re Q1, Q2, Q4, Q8, Q9
> > Thank you very much for clarifying, this level of comparison/delineation is what I have been missing. I understand that the 9-page-limit of a NeurIPS paper limits level of depth that can be spent on the related work; I would suggest, among other changes, to include a more comprehensive related work section in the appendix: The main points can then be made in the paper and a more detailed discussion can be included in the appendix (with 1-2 sentences in the main paper of what additional discussion can be found in the appendix).
> > I also want to encourage you to use the angle that you also used in the response to Q2: this is a straightforward idea, but making it work and scale well in practice is the non-trivial bit. This narrative also allows you to transition more easily between these two levels of abstraction in your paper as well as when describing your contributions: (i) relatively straightforward methodology; (ii) mathematical + implementation details. I think this also connects to Q10 and Q11: As a reader I prefer a very clear explanation/discussion of (i) before going to the details.
> > I also think that including a discussion along the lines of your replies to Q8 and Q9 in the paper and/or appendix would be very useful.
> >
> > ## Re Q10 + Q12
> > Thanks for providing more details/clarification in the paper on this.
> >
> > ## Re Q13
> > Thank you for the clarification; I'd propose to use the wording of "data subsampling" instead of "early stopping" as it seems cleaner to me and better describes what you're doing.
> >
> > ## Re Q11
> > I should have been clearer in my suggestion: I did not mean that $i$ was unclear. I was wondering whether it might make sense to explain a slightly simplified version of your approach on binary classification (hence without index $i$ as there is only one sigmoid output) to simplify the presentation and make it easier to follow; the added complexities of adding more outputs could then be discussed separately and presented in full in the Appendix. However, I also see that this is part of your contribution, so it might be harder to do. I was mainly trying to find ways to simplify the technical presentation so you could spend more time on providing intuition and analysis (e.g. to explain the overfitting issue in more detail, give more numbers to equations, etc).

---

> > > ### Author Response · Authors · 2022-08-09
> > > **Thank you very much!**
> > >
> > > We thank you very much for appreciating our response as well as being willing to adapt the score! We'll keep improving in the revision.
> > >
> > > Best,
> > > Authors

---

> > > > ### Comment · Reviewer_AKXi · 2022-08-10
> > > > **Thank you for the rebuttal**
> > > >
> > > > Thank you again for the clarifications and additional experiments that answered most of my questions.
> > > > I have raised the "soundness" and "presentation" score as well as the overall score and hope the authors address all of the promised changes (improvements in explanation + presentation; include additional explanations and results).
> > > > Overall, the paper is still borderline for me. For the level of novelty and contribution I expect a thorough analysis that explains the behaviour of the proposed algorithm/approximation in more detail. The two main areas that could still be explored and explained better:
> > > > * In-between uncertainty: Is the improved in-between uncertainty accidental? Can it purely be explained by "more approximate -> less certain -> more uncertainty"? Is this uncertainty in any way calibrated?
> > > > * Overfitting: How does the overfitting issue arise and why does data subsampling mediate it?

---

> > ### Comment · Reviewer_AKXi · 2022-08-04
> > **Follow up questions**
> >
> > I have some follow up questions that mainly relate to Q3 and would be curious to hear some more about these.
> >
> > First of all, thank you for the polite rebuttal on this; my pointed statement ("The paper ultimately discusses one particular approximation (Nystrom) to one particular covariance function (NTK) for one particular task") in the review was mainly due to the following
> > * this paper focuses on improving predictions though it doesn't state this clearly enough (in my opinion) in the abstract.
> > * the abstract says "we develop a Nystrom approximation to NTKs", which made me expect a wider scope of comparison (outside of pure LLA). I want to emphasise that I think focusing on ELLA is the right approach but suggest rephrasing in the abstract (potentially a sentence or two in the conclusion/outlook as to how your approach could be used in the NTK literature might be possible)
> >
> > My follow up questions are:
> > 1. Why does the regression performance behave the way it does in Fig 1, i.e. where does the added in-between uncertainty come from? ELLA is an approximation to a full GP model (i.e. the GP model before using Nystrom), which itself should be equivalent to a weight-space model, and a comparison to that is essential to understand the behaviour of ELLA. Could you discuss this a bit more? I assume the full GP model does not have any in-between uncertainty?
> > 2. Moreover, looking at the plots in Fig 1, one might reasonably argue that "LLA-KFAC" provides a better approximation to LLA than ELLA!?
> > 3. The overfitting behaviour still seems a bit concerning; if this is a problem with the Laplace approximation in general, it should be discussed as such (as this is a behaviour I have not seen discussed before). Your text makes it sound like this is a property of ELLA ("The Overfitting Issue of ELLA"). If the full GP/LLA model does not have this property, this would be concerning, as ELLA should be an approximation. Could you please elaborate further on this? I would argue that the data should shrink the uncertainty in areas where we observe a lot of data, while in data-sparse region the likelihood will not shrink the prior uncertainty too much (where closeness is defined through the NTK). I can see that data-subsampling is appealing from a practical perspective but it is also uncontrolled and should probably be discussed in more detail.
> > [a] Scalable Marginal Likelihood Estimation for Model Selection in Deep Learning (Immer et al 2021)

---

> > > ### Author Response · Authors · 2022-08-05
> > > **Further reply to Reviewer AKXi**
> > >
> > > We are excited to hear that you appreciate our rebuttal and consider raising your score. Here, we address your further questions and hope that you may find the response satisfactory. Feel free to comment if there are further questions.
> > >
> > > ### Regarding Q3
> > > Thanks for the valuable suggestions. We will revise the manuscript (especially the abstract/introduction) to clarify that this paper focuses on improving the probabilistic predictions of linearized Laplace approximation (LLA). We will also rephrase some sentences to make clear that in this paper, the developed Nystrom approximation to NTKs is mainly used for accelerating the computation of the predictive covariance of LLA, while it can also be easily applied to future works on the practical use of NTKs.
> > >
> > > ### Follow-up question \#1 \& \#2
> > > The full GP model mentioned in your question is the GP representation of LLA, whose covariance is defined in Eq 3. Eq 6 provides the reason for this point. Further, as shown in Sec 2.1, the GP representation of LLA and the weight-space representation of LLA are precisely equivalent, so the current comparison in Figure 1 is exactly what you want to see. By comparing the first and the second subplots, we observe that the full GP model has insufficient in-between uncertainty and state that ELLA offers better ‘in-between’ uncertainty than LLA. This is probably because ELLA relies on the Nystrom approximation of a $16\times 16$ matrix ($N=M=16$) with rank $5$ ($K=5$) (as detailed in Sec 5.3), so it should not be equivalent to the full GP model. Such inequivalence incurs this phenomenon. The underlying reason deserves further investigation – to the best of our knowledge, the concept of “in-between uncertainty” is usually discussed from a qualitative perspective (see references [14]), and there has not been a rigorous theoretical framework for discussing it.
> > >
> > > To check the goodness of the approximation to the original LLA, we quantify the discrepancies between the predictive distributions in Figure 1 using some metrics. Considering the predictive distribution for one test datum is a Gaussian distribution, we use the KL divergence between the Gaussians yielded by LLA and the method of concern as a proxy of the approximation error (averaged over a set of test points). The results are reported in the following table.
> > >
> > > |   |      ELLA      |  LLA-KFAC |  LLA-Diag  |  LLA*  |
> > > |----------|:-------------:|:-------------:|:-------------:|:-------------:|
> > > | ELLA |  0.83 | 0.35 | 1.71 | 2.44 |
> > > Table: Comparison on the approximation error to vanilla LLA, measured by the KL divergence between the predictive distribution.
> > >
> > > As mentioned by the reviewer, LLA-KFAC comes close to LLA. But, as argued, LLA might underestimate in-between uncertainty in this setting, so ELLA still seems to be a reliable approximation. We also highlight the higher approximation quality of ELLA than LLA-Diag and LLA*, as well as the higher scalability of ELLA than LLA-KFAC (see Figure 4c), which reflects that ELLA can strike a good trade-off between efficacy and efficiency. We thank the reviewer for his/her carefulness and will remove the arguments like "a closer approximation" in the revision.
> > >
> > > ### Follow-up question \#3
> > > Thanks for such a constructive question. To answer your question, we performed a study to check if LLA also suffers from the overfitting issue. As revealed by Table 1, LLA* is more suitable than LLA-Diag and LLA-KFAC when handling CIFAR-10 with ResNet architectures, so we experimented on LLA* here. We added the new results to Appendix B.1 of the manuscript (specifically, Figure 10) and updated the submission accordingly.
> > >
> > > As reflected by Figure 10 and Figure 2c, the overfitting issue of LLA generally exists. Nevertheless, we can see that properly tuning the prior precision can alleviate the overfitting of LLA$^*$ to some extent, which may be the reason why such an issue has not been reported by previous works. But it is also noteworthy that tuning the prior precision cannot fully resolve this issue.
> > >
> > > Given these, we have revised Sec 5.2 to clarify the generality of this issue and the necessity of introducing the early-stopping technique when applying linearized Laplace approximation to big data. Moreover, we argue that early-stopping, as a widely used trick in deep learning, is not pretty uncontrolled, given that we have provided tricks to construct surrogate validation data.
> > >
> > > Thank you again for raising such a constructive question. We will continually update the manuscript to make things clear and correct.

---

### Official Review · Reviewer_r6Sm · 2022-07-07

**Rating:** 7
**Confidence:** 4
**Soundness:** 3 good
**Presentation:** 3 good
**Contribution:** 4 excellent

**Summary:**

This work proposes a new, scalable variant of the (linearized) Laplace approximation (LLA) for inferring posterior distributions over the weights of deep learning models. Leveraging connections between LLA and neural tangent kernels (NTKs) -- namely that the LLA can essentially be viewed in function space as a Gaussian process with the NTK -- the paper describes how to speed up the LLA via a Nyström approximation to the NTK, effectively yielding an implicit sparse approximation to the LLA-GP predictive distribution. The work then elaborates on how to efficiently implement the proposed approach using automatic differentiation frameworks, and provides a theoretical analysis of the induced approximation error for the kernel. The resulting ELLA method is then empirically shown to outperform several baselines (both other LLA variants and a variational method) across a range of benchmarks -- even large-scale results with a Vision Transformer architecture on ImageNet are reported. It is also empirically found that ELLA (and LLA methods more generally) tends to "overfit" in some settings, which is proposed to be remedied by only using a subset of the data for estimating the covariance matrix, similar in spirit to early stopping methods often used in standard neural network training.

**Questions:**

- l. 234-235: in the footnote, you mention that "storing 2000 vectors of size $P$ is costly", but do you not just need to store $K=20$ instead of $M=2000$ vectors of size $P$ here (cf. l. 163-164)?
- l. 239-240: it would be good to understand why LLA-Diag and LLA-KFAC fail in this setting, given that previous works have been able to make them work fairly well, e.g. Daxberger et al. (2021); what have you tried to make these work and what is your intuition on why they fail?


**Limitations:**

Yes, the authors have adequately addressed this.

**Strengths And Weaknesses:**

**Summary of Review**

This work proposes a novel, theoretically-justified and clever variant of the Laplace approximation that is empirically shown to be performant and scalable to fairly modern deep learning settings, and that could therefore have significant impact by enabling practical uncertainty quantification for deep neural networks. Overall, there are many things that I liked about this paper; however, I also see some significant issues with the empirical evaluation (see details below). All in all, I am inclined to recommend acceptance of the manuscript, but would be more enthusiastic in my judgement if the authors can convincingly address the concerns raised.

**Strengths**
- The paper is overall well-written, easy-to-follow, and clearly-structured.
- The proposed ELLA method for accelerating LLA is novel, theoretically sound, and intuitively sensible; it is therefore a timely and welcome addition to the fast-growing Laplace toolbox for Bayesian deep learning.
- I appreciate the implementation-specific details in Section 3.4 including the code snippets in Algorithm 1-2, which make it fairly clear how one would practically implement ELLA (without needing to look at the actual Python code).
- The theoretical results on the approximation error induced by ELLA are nice-to-have (even though they appear to be somewhat straightforwardly inherited from previous analyses of Nyström-like kernel approximations).
- A main advantage of the proposed ELLA method is that a practitioner can actively control the performance vs. cost trade-off by choosing the hyperparameters $M$ and $K$, which is in contrast to most other methods (e.g. LLA-Diag and LLA-KFAC) that have a fixed cost and performance (however, as mentioned in the weaknesses below, this trade-off should be assessed more thoroughly in the empirical evaluation).
- The empirical evaluation demonstrates that ELLA can outperform relevant baselines on image-classification-based uncertainty calibration and out-of-distribution detection tasks; I particularly appreciate the inclusion of experiments on ImageNet, which is (unfortunately) still not so common in the Bayesian deep learning literature; I was especially impressed by the results on the recent Vision Transformer architecture, which promisingly demonstrates that the method can be applied in fairly modern deep learning settings (which, again, is not typically the case -- Bayesian methods normally lack behind significantly when it comes to adoption of advances in deep learning).
- I really liked the experiment showing that ELLA (and LLA methods more generally) have a tendency to "overfit" to larger datasets (in the sense that the test loss starts increasing again past a certain number of data points used for fitting the covariance matrix), demonstrating that LLA methods can be significantly sped-up by subsampling the data in some settings.

**Weaknesses**
- As all experiments seem to have been run with just a single random seed, the results do not come with any error bars, which makes it difficult to reason about the statistical significance of the reported conclusions; while I understand that repeating experiments for multiple random seeds linearly increases the required computational effort, reporting some sort of error statistic would significantly aid the credibility of the drawn conclusions; for the particularly expensive experiments (i.e. on ImageNet) I could perhaps accept lack of compute as an excuse (although even there, a minimum number of 3 seeds would be desirable), but on the cheaper experiments (i.e. on CIFAR-10) I would really expect some repetition of the experiments.
- The empirical comparison does not seem entirely fair, as different methods have different memory and compute requirements; in particular, for the hyperparameters chosen, ELLA seems to be more expensive than some of the other baselines considered (in terms of compute and/or memory cost, as e.g. also shown in Fig. 4c); therefore, it might be more appropriate and insightful to plot performances in 2D with memory and/or compute effort on the x-axis and performance on the y-axis, so that it becomes clear how the different methods can trade-off cost vs. performance; for ELLA, it would be great to then report performance for different values of $M$ and/or $K$ (i.e. different costs), yielding a Pareto curve; a practitioner can then choose the method/hyperparameters on the Pareto front that best fulfills their needs, i.e. which is either 1) as cheap as possible for a given desired performance X, or 2) as performant as possible for a given desired cost Y; without such a plot it is difficult to draw conclusions on how superior ELLA really is -- although ELLA's ability to cater to a wide range of performance/cost trade-offs by sweeping over $M$ and $K$ should already make it look better than most of the baselines; you already show several plots that are related in spirit (Fig 2. a-b, Fig. 4 b-c), but I believe the specific kind of plot I described would be much more insightful.
- For the ImageNet experiments, I would expect that at least the last-layer KFAC variant should be feasible to run and not yield an OOM error (with either a ResNet or ViT) -- e.g. the Eschenhagen et al. (2021) paper you cite seems to have managed to run this Laplace variant on ImageNet with a ResNet; I think it would significantly strengthen the paper to at least have this one Laplace baseline to compare with; if this is really an issue with the Laplace library, I would encourage you to reach out to the authors / raise an issue on the GitHub repo -- I also had issues with this library in the past, and in my experience, the authors are typically happy to help and fairly quick to respond.
- The proposed approach can be viewed as limited in novelty in the sense that it effectively combines the LLA with a Nyström approximation to the NTK, which both have been studied (fairly extensively) before. That being said, I think the idea is pretty neat (even though somewhat obvious in hindsight) and appreciate that the authors made the effort of actually getting this to work, so I don't view this to be a major issue.

**Minor issues**
- l. 226-227: to back up the claim that "ELLA delivers a closer approximation to LLA" it might be worth explicitly quantifying the discrepancies between the predictive distributions using some metric; qualitatively (i.e. by just looking at the plots), it seems like LLA-KFAC also comes pretty close (but I agree that even LLA might underestimate in-between uncertainty in this setting, so the ELLA fit could be more desirable).
- Generally, it would be useful to have short take-away messages in the captions of each table/figure for clarity and convenience.

---

> ### Author Response · Authors · 2022-08-01
> **Response (part II/II)**
>
>
> |   |  4 |  8 |  16 |  32 |  64 |  128 |
> |----------|:------:|:------:|:------:|:------:|:------:|:------:|
> | 4 | 0.2689  |   |   |   |   |   |<!--   |   |   |   |-->
> | 8 |  0.2690 | 0.2561  |   |   |   |   |<!--   |   |   |   |-->
> | 16 | 0.2660  |  0.2548 |  0.2392 |   |   |   | <!--  |   |   |   |-->
> | 32 |  0.2655 | 0.2541  | 0.2395  | 0.2318  |   |   |<!--   |   |   |   |-->
> | 64 | 0.2672  |  0.2527 |  0.2398 | 0.2312  | 0.2299  |   | <!--  |   |   |   |-->
> | 128 |  0.2674 | 0.2540  |  0.2383 | 0.2310  |  0.2298 | 0.2294  | <!--  |   |   |   |-->
> | 256 |  0.2679 |  0.2545 |  0.2382 |  0.2310 | 0.2300  |  0.2294 |<!-- 0.2322  |   |   |   |-->
> | 512 |  0.2678 | 0.2542  |  0.2376 |  0.2315 |   0.2292|  0.2288 | <!--0.2320  | 0.2337  |   |   |-->
> | 1024 |  0.2674 |  0.2545 |  0.2385 |  0.2314 |  0.2295 |  0.2292 | <!--0.2316  | 0.2333  | 0.2350  |   |-->
> | 2000 |  0.2673 | 0.2551  | 0.2380  | 0.2308  |   0.2301| 0.2295  | <!--0.2319  |  0.2334 |  0.2350 |   |-->
> Table: Test NLL varies w.r.t. $M$ (vertical axis) and $K$ (horizontal axis) ($M\geq K$) on CIFAR-10 with ResNet-20 architecture
>
>
> ### Q4: Regarding the ImageNet experiments
>
> We launch a new experiment using LLA*-KFAC with ResNet-18 architecture on ImageNet. LLA\*-KFAC is arguably more flexible and expressive than LLA\*-Diag. The hardware is an NVIDIA A100 GPU with 40G memory. The results as well as those of ELLA are presented in the following table.
>
>
> |   |      Acc.(%)      |  NLL |  ECE  |
> |----------|:-------------:|:-------------:|:-------------:|
> | ELLA |  **69.8** | **1.243** | 0.016 |
> | LLA*-FKAC |    69.6   |   1.246 | **0.015** |
> Table: Comparison on test accuracy (%), NLL, and ECE on ImageNet with ResNet-18 architecture.
>
> As shown, ELLA demonstrates better accuracy and NLL but slightly worse ECE than LLA\*-KFAC. This may be attributed to that we explicitly tune the prior precision for LLA*-KFAC but not for ELLA (in all experiments; see the call of `la.optimize_prior_precision` in the code). Considering LLA\*-KFAC requires higher memory consumption (at least in the implementation of the Laplace library; see the Issue panel of its GitHub repo) and longer fitting (as ELLA easily embraces early stopping), ELLA is still a better choice in ImageNet settings. We will add the ImageNet results with more architectures in the revision.
>
> ### Q5: Regarding novelty
>
> We thank the reviewer for acknowledging that the idea is neat and our effort is meaningful. Thank you!
>
> ### Q6: Regarding "ELLA delivers a closer approximation to LLA"
>
> We agree with the reviewer that it is better to quantify the discrepancies between the predictive distributions using some metric. Here, considering the predictive distribution for one test datum is a Gaussian distribution, we use the KL divergence between the Gaussians yielded by LLA and the method of concern as a proxy of the approximation error (averaged over a set of test points). The results are reported in the following table.
>
> |   |      ELLA      |  LLA-KFAC |  LLA-Diag  |  LLA*  |
> |----------|:-------------:|:-------------:|:-------------:|:-------------:|
> | ELLA |  0.83 | 0.35 | 1.71 | 2.44 |
> Table: Comparison on the approximation error to vanilla LLA, measured by the KL divergence between the predictive distribution.
>
> As mentioned by the reviewer, LLA-KFAC comes pretty close to LLA. But ``even LLA might underestimate in-between uncertainty in this setting'', so ELLA seems to be a *more reliable* approximation than LLA-KFAC. We thank the reviewer for his/her carefulness and will remove the arguments like "a closer approximation" in the revision.
>
> ### Q7: Include short take-away messages in the captions of each table/figure
> Thanks for the suggestion. We will revise the paper accordingly.
>
> ### Q8: Regarding "storing 2000 vectors of size is costly"
> Thanks for the question. Note that in the fitting phase of ELLA (building $\psi$), we need to obtain the top-$K$ eigenpairs of a $M \times M$ matrix and the $M \times M$ matrix corresponding to the inner products between $M$ vectors of size $P$ (see L144). In practice, $M=2000$, $P>10^6$, and $K=20$, so we cannot store all the $M$ vectors of size $P$ to compute the $M \times M$ matrix. We opt to trade time for memory and compute the elements of the $M \times M$ matrix one by one (or one block by one block) with temporary vectors of size $P$. After this, we only need to store $K=20$ vectors (i.e., $v_i$) for testing.
>
>
> ### Q9: Why LLA-Diag and LLA-KFAC fail
> Currently, we have not figured out the reason why LLA-Diag and LLA-KFAC fail. This phenomenon is also reported by [9]. We have enabled the tuning of the prior precision for these two methods, and the experimental settings are the same as LLA* baselines. We do not know if this is due to bugs in the Laplace library or some un-identified subtleties. We will later contact the authors of the Laplace library to make it clear.

---

> > ### Comment · Reviewer_r6Sm · 2022-08-08
> > **Thank you for your response**
> >
> > Thanks a lot for your thorough and helpful response, which clarifies most of the concerns I had. After also having read the other reviews and corresponding author responses, I am happy to increase my score to 7 and recommend acceptance more enthusiastically. I think this is a great paper that will be useful to many people. While I agree with some of the remaining concerns that the other reviewers have, I personally think that a paper does not need to be perfect in all aspects, and, in my opinion, this paper meets the bar for acceptance despite the remaining issues.

---

> > > ### Author Response · Authors · 2022-08-08
> > > **Thank you**
> > >
> > > Thanks a lot for your support! We will continually improve the manuscript and try our best to address the remaining issues.

---

> ### Author Response · Authors · 2022-08-01
> **Response (part I/II)**
>
> We appreciate the positive review and address the detailed comments below.
>
> ### Q1: Regarding error bars
>
> Thanks for the nice comments. Following the suggestion, we repeat the experiments in Table 1 for 4 other runs with random seeds and update the table as follows:
>
>
>
> |    ||                ResNet-20  |||   ResNet-32                  |||  ResNet-44                  ||| ResNet-56                  ||
> | :------------- | -----------: | -------------: | ----------: |  -----------: | -------------: | ---------: |  -----------: | -------------: | ----------: | -----------: | -------------: | ----------: |
> |            |   Acc. | NLL | ECE | Acc. | NLL | ECE | Acc. | NLL | ECE |  Acc. | NLL | ECE |
> |*ELLA*            |   0.9254±0.0003 | 0.2330±0.0017 |**0.0094**±0.0027|0.9350±0.0003 | **0.2151**±0.0011|**0.0077**±0.0013|0.9392±0.0004|**0.2035**±0.0008|**0.0074**±0.0019| 0.9440±0.0003|**0.1865**±0.0011|**0.0073**±0.0007|
> | *MAP*            |   0.9260 | 0.2815 |0.0390 |0.9353 | 0.2921|0.0408|0.9401|0.2753|0.0385| 0.9437|0.2523|0.0374|
> | *MFVI-BF*            |   0.9267±0.0010 | **0.2314**±0.0010 |0.0162±0.0012 |0.9354±0.0005 | 0.2222±0.0012|0.0197±0.0004|0.9394±0.0007|0.2056±0.0025|0.0178±0.0009|0.9440±0.0008|0.1880±0.0009|0.0159±0.0004|
> | *LLA**            |   0.9257±0.0001 | 0.2689±0.0001 |0.0339±0.0001 |0.9351±0.0002 | 0.2589±0.0003|0.0329±0.0004|0.9398±0.0002|0.2365±0.0006|0.0276±0.0003| 0.9436±0.0002|0.2132±0.0002|0.0219±0.0003|
> | *LLA\*-KFAC*            |   0.9257±0.0001 | 0.2705±0.0001 |0.0345±0.0002 |0.9351±0.0002 | 0.2595±0.0003|0.0331±0.0003|0.9398±0.0002|0.2318±0.0004|0.0279±0.0003| 0.9436±0.0002|0.2017±0.0002|0.0235±0.0005|
> | *LLA-Diag*            |   0.9215±0.0015 | 0.7284±0.0014 |0.4037±0.0021 |0.9266±0.0016 | 0.7547±0.0028|0.4300±0.0013|0.9275±0.0009|0.7781±0.0015|0.4451±0.0012| 0.9289±0.0009|0.8426±0.0033|0.4803±0.0015|
> | *LLA-KFAC*            |   0.9202±0.0010 | 0.8515±0.0014 |0.4668±0.0012 |0.9184±0.0013 | 1.0269±0.0028|0.5467±0.0012|0.9136±0.0009|1.0913±0.0014|0.5660±0.0010| 0.9440±0.0008|0.1880±0.0009|0.0159±0.0004|
>
>
>
> Furthermore, we repeat the ImageNet experiments for 2 other runs and update Table 2 as follows:
>
>
> |    ||                ResNet-18  |||   ResNet-34                 |||  ResNet-50                  ||
> | :------------- | -----------: | -------------: | ----------: |  -----------: | -------------: | ---------: |  -----------: | -------------: | ----------: |
> |            |   Acc. | NLL | ECE | Acc. | NLL | ECE | Acc. | NLL | ECE |
> |*ELLA*            |   0.6981±0.0002 | 1.2431±0.0001 |**0.0153**±0.0007|0.7332±0.0003 | 1.0724±0.0015|**0.0177**±0.0031|**0.7617**±0.0001|0.9479±0.0001|**0.0179**±0.0004|
> | *MAP*            |   0.6976 | 1.2469 |0.0263 |0.7330 | 1.0810|0.0354|0.7615|0.9616|0.0369|
> | *MFVI-BF*            |   **0.7027**±0.0001 | **1.2177**±0.0019 |0.0423±0.0012 |**0.7374**±0.0009 | **1.0432**±0.0018|0.0327±0.0008|0.7614±0.0004|**0.9452**±0.0006|0.0298±0.0022|
>
> The results are in good agreement with existing results in the manuscript and the drawn conclusions still hold. Feel free to comment if there are further questions.
>
> ### Q2: Regarding memory and compute requirements
>
> Thanks for the comments. We make the following clarification.
>
> On the one hand, the detailed memory/compute consumption relies on specific model architectures. As discussed in L162-167, $K$ can be a good proxy for the memory/compute consumption in the test phase of ELLA. Therefore, Figure 4b has provided the plot with memory/compute effort on the x-axis and performance on the y-axis. On the other hand, as agreed by the reviewer, the other methods (e.g., LLA-Diag and LLA-KFAC) cannot ``actively control the performance vs. cost trade-off''. And the reported Figure 4c and Table 1 jointly present the inter-method comparison on the axes of cost and performance. Therefore, we argue that these results do not reflect an unfair comparison but ELLA's specific advantages and flexibility.
>
> By the way, we respectfully point out that ELLA is more expensive than LLA* and LLA\*-KFAC in Figure 4c because LLA* and LLA\*-KFAC are both variants of *last-layer* Laplace. Only a low-dimensional parameter distribution is handled in these cases, so they are fast. Comparison between ELLA and LLA-Diag/LLA-KFAC is fairer.
>
>
> ### Q3: Performance for different values of $M$ and $K$
>
> We appreciate this kind suggestion. We perform systematic experiments in this spirit and present the results in the following table. The experiment for $K\geq256$ is time-consuming so we have not included the corresponding results here. We find that under the same $K$, a larger $M$ can bring closer approximation (see Figure 2) but it does not necessarily lead to a better test NLL. We will add these to the revision.

---

### Official Review · Reviewer_2eHY · 2022-07-11

**Rating:** 6
**Confidence:** 4
**Soundness:** 3 good
**Presentation:** 3 good
**Contribution:** 2 fair

**Summary:**

The paper proposes ELLA, a Nyström approximation, as an alternative posterior (predictive) approximation for linearized Laplace. Their method uses the NTK-based formulation of linearized Laplace and improves over other common Laplace approximations in terms of quality of the predictive while maintaining a relatively low computational complexity.

**Questions:**

- ll 31 says that LLA has a more benign covariance. Can you elaborate on what that means or in what way it is benign?
- Figure 1: how comes that ELLA gives better "in-between" uncertainty? Are the hyperparameters equivalent to the LLA method? Generally, ELLA approximates the LLA variant so it is unclear why/how it is better. This would be worth investigating more.
- Could you elaborate on the simplification in Equation (6), for example, by putting a detailed derivation in the appendix? It is not clear to me where the equivalence comes from.
- Regarding computational overhead (ll. 162 ff): Can you relate this to the prevelant alternative of KFAC-LLA? It would be interesting to know the complexity of computing the posterior predictive for an unseen data point after inference.
- The paper at hand only discusses the posterior predictive. Have the authors considered using the Nyström approximation for marginal likelihoods as well. In [*scalable marginal likelihood estimation for model selection in deep learning*](https://arxiv.org/abs/2104.04975), the NTK variant for linearized Laplace was used but without further approximations. It would be interesting to experiment if Nyström can provide an alternative scalable method. Potentially, this could alleviate overfitting in Figure 2c) by selecting the number of bases $K$?
- Figure 2 is hard to parse since the colorbars are unclear. What is the differene between Nyström and ELLA there in particular?
- Table 2 has no error bars but bold figures. Which policy is used to make numbers bold? Did the authors run for several seeds? Same Q for Table 3.
- Regarding the overfitting issue of ELLA: to what extent can changing the prior fix such overfitting issue? For example, the prior precision is often changed after training to improve the performance of LA [6]. Is this an option here, too?

**Limitations:**

There are no immediate societal impacts of the present work. Computational limitations are discussed in the paper, for example, the authors discuss the issue of overfitting present in their method.

**Strengths And Weaknesses:**

## Strengths
- Can reduce the cubic complexity in data points $N$ or parameters $P$ of linearized Laplace in neural networks down to much lower complexity that can be controlled by the number of samples used for Nyström. This way, a performance-accuracy trade-off can be made.
- Performance of the proposed posterior predictive improves over cheap last-layer Laplace and is better than the full-network variants.
- Interesting and open discussion of the overfitting issue of the proposed method ELLA.
- Exhaustive experiments with ablations on the number of factors $K$ and several metrics of interest. Overall convinces that the proposed method is a valuable addition to the family of Laplace approximations.

## Weaknesses
- Only posterior predictive is discussed but not marginal likelihood estimation. Since the proposed method is rather simple ("take linearized Laplace in its kernel variant and apply Nyström, which is well-known from GP literature"), it would greatly strengthen the paper to at least show not only predictives but also the marginal likelihood, which should in principle be easy to add.
- Misunderstanding of relationship between linearized Laplace and GGN. ll 3-6 and 30-32: linearized Laplace and Laplace with GGN are equivalent as shown in reference [22]. The equivalent variant with (neural tangent) kernels has been used before as well in [22,28] but without Nyström approximations. ll 92-99 discusses the scalability issues only of the parametric LLA which requires approximations to the $P \times P$ GGN. The problem that is tackled in the present paper, however, is the approximation of the equivalent $N \times N$ kernel, which is for example defined. Clearly, $N \times N$ is still intractable so the proposed Nyström approximation is well justified.
- ELLA does not seem to approximate full LA, which it should in theory. Figure 1: discrepancy of ELLA and LLA, it seems like ELLA does not get very close to LLA in Figure 1 and there is no explanation as to why that would be. In line 50, it is said though that the approximation becomes accurate.
- Relation to prior work could be pointed out more clearly. For example, it is unclear to what extent the Theorems are novel/original or if they follow in a straightforward way from previous theoretical contributions.

---

> ### Author Response · Authors · 2022-08-01
> **Response (part II/II)**
>
>
>
> ### Q9: Relate this to the prevalent alternative of KFAC-LLA? The complexity of computing the posterior predictive for an unseen data point?
>
> In the fitting phase of KFAC-LLA, we need to iterate over the training set to estimate the KFAC approximation of the GGN matrix, where each iteration entails one forward-backward pass. Then we invert the block-diagonal matrix to obtain the Gaussian covariance $\Sigma$. When testing on one datapoint, KFAC-LLA performs one forward pass and $C$ backward passes to compute and store the large $C\times P$ Jacobian matrix $J$, then computes the predictive covariance $J\Sigma J^\top$ ($C$ is the number of classes). On the other hand, the evaluation of ELLA requires performing $K$ fwAD forward passes without storing Jacobian matrices (clarified in L166). When $K$ is not big, the evaluation of ELLA is arguably faster than KFAC-LLA. The empirical comparison in Figure 4c confirms this.
>
>
> ### Q10: Figure 2 is hard to parse.
>
> Thanks for the suggestion, and we will revise the color bars to make Figure 2 more readable. $\epsilon_\text{Nyström}$ and $\epsilon_\text{ELLA}$ have been elaborated in the corresponding text (L197-198). Specifically, $\epsilon_\text{Nyström}$ measures the error of the Nyström approximation ($\mathcal{E}'$ in Theorem 1), and $\epsilon_\text{ELLA}$ measures the gap between the predictive covariance between $ELLA$ and $LLA$.
>
> ### Q11: Regarding Table 2
>
> We highlight the best accuracy $\uparrow$, NLL $\downarrow$, and ECE $\downarrow$ in bold. $\downarrow$ means the lower, the better, and $\uparrow$ means the higher, the better. For Table 1, as the accuracies of most methods are close, we didn't highlight the best accuracies.
>
> We repeat the experiments in Table 2 for 2 other runs with random seeds and update the tables as follows:
>
> |    ||                ResNet-18  |||   ResNet-34                 |||  ResNet-50                  ||
> | ------------- | -----------: | -------------: | ----------: |  -----------: | -------------: | ---------: |  -----------: | -------------: | ----------: |
> |            |   Acc. | NLL | ECE | Acc. | NLL | ECE | Acc. | NLL | ECE |
> |*ELLA*            |   0.6981±0.0002 | 1.2431±0.0001 |**0.0153**±0.0007|0.7332±0.0003 | 1.0724±0.0015|**0.0177**±0.0031|**0.7617**±0.0001|0.9479±0.0001|**0.0179**±0.0004|
> | *MAP*            |   0.6976 | 1.2469 |0.0263 |0.7330 | 1.0810|0.0354|0.7615|0.9616|0.0369|
> | *MFVI-BF*            |   **0.7027**±0.0001 | **1.2177**±0.0019 |0.0423±0.0012 |**0.7374**±0.0009 | **1.0432**±0.0018|0.0327±0.0008|0.7614±0.0004|**0.9452**±0.0006|0.0298±0.0022|
>
> The results are in good agreement with existing results in the manuscript and the drawn conclusions still hold. Find the updated Table 1 in [the reply to the first question of Reviewer r6Sm](https://openreview.net/forum?id=jftNpltMgz&noteId=AO6d4uNbqZx). Feel free to comment if there are further questions.
>
>
>
> ### Q12: To what extent can changing the prior fix such an overfitting issue?
>
> Yes, changing the prior is a viable option to improve the performance of LA [6,7] and fix such an overfitting issue. By changing the prior, the strength of the regularization is adjusted; by early stopping, the strength of data evidence is weakened. So the two tricks may ideally lead to similar outcomes, and in practice, they both rely on extra validation data (clarified in L219). Nevertheless, as stated in L221, an additional benefit of early stopping is that it significantly reduces ELLA's time cost for handling large data (e.g., ImageNet).

---

> > ### Comment · Reviewer_2eHY · 2022-08-08
> > **Discussion**
> >
> > Thank you for the detailed response. I do think this is a good paper but there are some remaining unclear outcomes that might need further investigation such as the role of the prior precision on ELLA, overfitting of ELLA, and the relationship between the two. It is also questionable to what extent the better in-between uncertainty is really a feature of the method rather than an issue and I do think this deserves further investigation as pointed out by reviewer AKXi. I am unsure if all these things can be fixed for a camera-ready version, which is why I currently tend to keep my score.

---

> > > ### Author Response · Authors · 2022-08-08
> > > **Reply**
> > >
> > > We appreciate the further comment.
> > >
> > > In fact, in the reply to the *Follow-up question #3* [here](https://openreview.net/forum?id=jftNpltMgz&noteId=EExH-iAKfcM), we added new experiments to make clear that (1) the overfitting issue of LLA methods generally exists; (2) Properly tuning the prior precision can alleviate the overfitting of LLA to some extent, which may be the reason why such an issue has not been reported by previous works. But also of note that tuning the prior precision cannot fully resolve the overfitting issue and it is desirable to deploy early-stopping. See Figures 10 & 2 of the updated manuscript for more details. These findings apply to ELLA as well because ELLA is a flexible approximation to LLA.
> > >
> > >
> > > Besides, we will experiment on tasks like "gapped UCI regression" [14] to check the generality of the "better" in-between uncertainty of ELLA. We will also carefully revise the arguments regarding Figure 1.
> > >
> > > Thanks!

---

> ### Author Response · Authors · 2022-08-01
> **Response (part I/II)**
>
> Thanks for the positive reviews and valuable comments. We address the detailed concerns below.
>
> ### Q1: The paper at hand only discusses the posterior predictive. Have the authors considered using the Nyström approximation for marginal likelihoods as well?
>
> We thank the reviewer for the suggestion. In fact, we have pointed out such a valuable future work in the conclusion section (see L284). We clarify that this is not a *limitation* of our work but an interesting and easy extension. Due to limited energy, we are currently focusing on the *typical* task -- improving the posterior predictive of Laplace approximation (previous works like [50, 22] also focus on this). The contributions of this paper on this point are unignorable and are appreciated by you and Reviewer r6Sm. Considering the tight rebuttal period, we will try to include initial results on using marginal likelihoods for model selection in the revision (because the experimental settings for model selection are significantly different from those in the current manuscript).
>
> By the way, we agree with the reviewer that by inspecting the marginal likelihood, we can choose a *good* $K$ to alleviate overfitting in Figure 2c. In fact, cross-validation (used in this paper) and marginal likelihoods are two canonical tools for model selection.
>
>
> ### Q2: Misunderstanding of the relationship between linearized Laplace and GGN
>
> Thanks. We make the following clarification. According to previous work [50,30,14], in our manuscript, (a) Laplace with GGN refers to replacing the Hessian in vanilla Laplace with GGN, but the prediction is still made with model $p(y|f(x,\theta))$ (i.e., the BNN predictive in [22]); (b) Linearized Laplace refers to using the GGN approximation to Hessian and using the linearized model $p(y|f_{lin}^{\hat{\theta}}(x,\theta))$ for prediction (i.e., the GLM predictive in [22]). [22] states that GGN approximation turns the model locally from a BNN into a GLM, which means that the GLM predictive is more proper than BNN predictive when employing GNN approximation and does *not* imply that linearized Laplace and Laplace with GGN are equivalent. We will revise the paper accordingly.
>
>
> ### Q3: Discussion on the scalability issues
>
> In L92-99, we discuss the scalability issue of manipulating the huge $P\times P$ GGN matrix. To solve it, we explore the GP view of LLA and observe that the scalability issue turns into a dual one where we are confronted with the $NC \times NC$ gram matrix (described in L111-116). We then naturally apply the Nyström method for kernel approximation for speedup. The whole logic chain is well justified.
>
>
> ### Q4: ELLA does not seem to approximate full LA in Figure 1
>
> As detailed in Sec 5.3, for ELLA in Figure 1, we set $K=5$, which is substantially smaller than $M=N=16$, so ELLA is *low-rank* and cannot approximate full LLA quite precisely. But it has already surpassed the baselines like LLA-Diag and LLA*. In L50, we state that as the Nyström approximation becomes accurate, the approximation error between the predictive of ELLA and that of vanilla LLA decreases, which is reflected by Theorem 1 and Figure 2(a)-(b).
>
> ### Q5: Relation to prior work could be pointed out more clearly
>
> To the best of our knowledge, Theorem 1 is novel and derived by ourselves. Theorem 2 is adapted from Drineas and Mahoney [12]     \(stated in the manuscript). Corollary 1 is straightforward to obtain. We will make these more apparent in the next version.
>
>
> ### Q6: Regarding that "LLA has a more benign covariance"
>
> By saying "a more benign covariance", we mean the predictive covariance in function space, or more precisely, predictive uncertainty. [14] provides support for this. We will revise the manuscript to make it clear.
>
>
> ### Q7: How comes that ELLA gives better "in-between" uncertainty? Are the hyperparameters equivalent to the LLA method?
>
> We affirm that the hyperparameters of ELLA are equivalent to those of the LLA methods except for some dedicated ones like $M$ and $K$. We point out that LLA is likely to underestimate the in-between uncertainty in this setting (we test an NN-GP model on this problem and observe higher in-between uncertainty). As detailed in Sec 5.3, ELLA relies on the Nystrom approximation of a $16\times 16$ matrix ($N=M=16$) with rank $5$ ($K=5$), so it should not be equivalent to the original LLA. Such inequivalence incurs better in-between uncertainty for ELLA, and the underlying reason deserves further investigation. We will further check the generality of this phenomenon and carefully revise the arguments in the revision.
>
>
> ### Q8: Could you elaborate on the simplification in Equation (6)
>
> Appendix A.2 has already provided the details.

---

### Author Response · Authors · 2022-08-01
**Thanks to all the reviewers for their efforts**

We thank all reviewers for their efforts in providing insightful comments and constructive feedback. We are encouraged that reviewers recognize that our work is a valuable, timely, and welcome addition to the family of Laplace approximations [R1, R2], and it enables the desirable performance-accuracy trade-off for Bayesian deep learning [R1, R2] and provides an interesting and open discussion of the overfitting issue [R1, R2]. In the following, we address reviewers’ comments point by point.

---

### Author Response · Authors · 2022-08-08
**Look forward to further feedback**

Dear reviewers,

Thank you for your thorough review. We are looking forward to your reply. If you have any further concerns or requests, we can have a chance to address them in the author-reviewer discussion period. If all your concerns have been resolved, it is much appreciated if you may raise the rating of our work.

Best,

Authors

---

### Meta-Review · Area_Chair_C3VP · 2022-08-27

**Recommendation:** Accept
**Confidence:** Less certain

**Metareview:**

This paper introduces an approach to accelerating linearized Laplace approximations to Bayesian neural network posteriors, particularly considering prediction tasks, by performing a Nyström approximation to the neural tangent kernel.

The reviewers all recommended acceptance, eventually — one reviewer was initially quite critical but revised after a rather extensive discussion with the authors revised to a borderline accept. In particular some additional experiments analyzing overfitting in these models, in general, were appreciated.

While this is perhaps borderline on the scores (5,6,7), given the overall quality of the work and the extent to which this was updated and improved during the rebuttal period, I would recommend acceptance.

**Award:**

No

---

### Decision · Program_Chairs · 2022-09-14

Accept